Quantum-effective exact multiple patterns matching algorithms for biological sequences

http://orcid.org/0000-0002-2107-2121 Soni Kapil Kumar prof.kapilsoni@gmail.com
Rasool Akhtar
Department of Computer Science and Engineering, Maulana Azad National Institute of Technology , Bhopal, Madhya Pradesh , India
Bhattacharyya Siddhartha
Electronic publication date: 2022 May 12
Publication date: 2022
Volume: 8
Electronic Location ID: e957
Received 2021 Jul 29; Accepted 2022 Apr 1
Copyright: © 2022 Soni and Rasool
Copyright year: 2022
Copyright holder: Soni and Rasool
License: This is an open access article distributed under the terms of the Creative Commons Attribution License, which permits unrestricted use, distribution, reproduction and adaptation in any medium and for any purpose provided that it is properly attributed. For attribution, the original author(s), title, publication source (PeerJ Computer Science) and either DOI or URL of the article must be cited.
License URL: https://creativecommons.org/licenses/by/4.0/

Keywords: Quantum algorithms, Biological sequences, Grover’s quantum search, Quantum memory, Quantum exact multiple pattern matching

Funding: The authors received no funding for this work.

==============================
This article presents efficient quantum solutions for exact multiple pattern matching to process the biological sequences. The classical solution takes Ο(mN) time for matching m patterns over N sized text database. The quantum search mechanism is a core for pattern matching, as this reduces time complexity and achieves computational speedup. Few quantum methods are available for multiple pattern matching, which executes search oracle for each pattern in successive iterations. Such solutions are likely acceptable because of classical equivalent quantum designs. However, these methods are constrained with the inclusion of multiplicative factor m in their complexities. An optimal quantum design is to execute multiple search oracle in parallel on the quantum processing unit with a single-core that completely removes the multiplicative factor m, however, this method is impractical to design. We have no effective quantum solutions to process multiple patterns at present. Therefore, we propose quantum algorithms using quantum processing unit with C quantum cores working on shared quantum memory. This quantum parallel design would be effective for searching all t exact occurrences of each pattern. To our knowledge, no attempts have been made to design multiple pattern matching algorithms on quantum multicore processor. Thus, some quantum remarkable exact single pattern matching algorithms are enhanced here with their equivalent versions, namely enhanced quantum memory processing based exact algorithm and enhanced quantum-based combined exact algorithm for multiple pattern matching. Our quantum solutions find all t exact occurrences of each pattern inside the biological sequence in O((m/C)N) and O((m/C)t) time complexities. This article shows the hybrid simulation of quantum algorithms to validate quantum solutions. Our theoretical–experimental results justify the significant improvements that these algorithms outperform over the existing classical solutions and are proven effective in quantum counterparts.

Introduction

The exact multiple pattern matching problem is to find a bijective mapping for m patterns within the text sequence database. Searching for the multiple string patterns would be more practical while processing large biological sequence databases (Basel, 2006; Neamatollahi, 2020). The search of multiple nucleotides or amino acid patterns is necessary within the genome, protein and other biological sequences for a significant purpose (Charalampos, Panagiotis & Konstantinos, 2011). For example, we know that proteogenomics mapping uses proteomics data for DNA or genome annotation. This mapping matches peptide or protein patterns within the proteomics data through the mass spectrometry analysis against the target genome for identifying all locations of genes with coding regions (Choo, 2006; Fredriksson, 2009). Therefore, this processing demands a compatible and efficient solution to search for multiple patterns belonging to P={P1,.,Pk,.,Pm} with |P|=m. Each pattern Pk (1≤k≤m) of independent length Mk=[0 to Mk−1] is searched within the large-sized text sequence T of length N=[0 to N−1]. Both M and N belongs to the alphabet set Σ such that N≫M, and t number of pattern occurrences is possible to search between index positions 0 and N − M. Now, specific to biological sequence processing, we usually prefer these m patterns with the same length. Certainly, the set P contains multiple patterns, although, the restricted singleton set |P|=1 allows us to search P as a single pattern (Charalampos, Panagiotis & Konstantinos, 2011; Faro & Lecroq, 2013; Zhang et al., 2015; Hendrian et al., 2019; Hakak & Kamsin, 2019).

The biological sequence database contains N sized text with exponential factors of gigabytes, terabytes or more. For single pattern matching, the classical solution scans these databases in directional sequence on the main memory (Sheik, Aggarwal & Anindya Poddar, 2004; Kalsi, Peltola & Tarhio, 2008; Rivals, Salmela & Tarhio, 2011). The search time is still bound to O(N) or the complete scan of text to find all the t occurrences of P(|P|=1); however, for the exponentially large value of N, the problem is computationally hard. Thus, we clarify that the time of pattern search increases in proportion to the size of text database, so fast searching techniques are expected. A classical method for the multiple pattern matching takes O(mN) time complexity due to repeated scanning of N sized text database for m patterns (Fredriksson, 2009; Charalampos, Panagiotis & Konstantinos, 2011). In contrast, the quantum search takes O(N) time (Nielsen & Chuang, 2010); therefore, a quadratic speedup is possible, and such acceleration is expected in quantum pattern matching (Soni & Rasool, 2020). Since the existence of problem, solutions have been suggesting through modified algorithms.

The objective is to suggest effective pattern matching algorithm with better performance than others and to set itself as a benchmark solution. We seek technology-based solutions; therefore, effective quantum-based algorithms are expected for multiple pattern matching. Some quantum-based exact single pattern matching algorithms are enhanced here for their equivalent multiple pattern matching versions (Soni & Rasool, 2021). Our methods remove existing multiple pattern matching constraints (Soni & Malviya, 2021) and realize the effective quantum-based solutions by scanning the text database in the uniform superposition of quantum memory (QMEM) (Giovannetti, Lloyd & Maccone, 2008; Nielsen & Chuang, 2010). Therefore, based on the advantage of quantum processing unit (QPU) having C quantum cores (1≤c≤C) (Metodi, 2011; Lin et al., 2013; Fu et al., 2016; Britt, 2017; Brandl, 2017), we propose our algorithms to match m patterns using the quantum-exact match (QEM) circuit (Sena Oliveira, Benicio Melo de Sousa & Viana Ramos, 2007; Soni & Rasool, 2021) and quantum Grover’s search operator (GSO) mechanism (Nielsen & Chuang, 2010; Chakrabarty, Khan & Singh, 2017).

Significance of processing biological sequences

For processing the biological sequence databases, exact matches are always preferred with accurate matching outcomes. The nucleotide and amino acid patterns are used to locate within genome, protein and other biological sequences for different purposes (Jiang, Zhang & Zhang, 2013; Singh, 2015). The size of DNA or RNA alphabet set is |Σ|=4, and coded adjacent triplet of nucleotide characters which forms amino acid with the set size |Σ|=20. The biological sequence databases are excessively large, so multiple string patterns should be effectively processed. Multiple pattern matching aims to identify all locations of m patterns within the sequence databases in a single scan. The searching of DNA pattern within nucleotide sequence helps us to identify, compare and align the sequences as well as to analyze mutations (Faro & Lecroq, 2009; Tahir, Sardaraz & Ikram, 2017; Raja & Srinivasulu Reddy, 2019). However, different nucleotides can code to similar proteins, so protein databases are searched for similarity checks.

An exact multiple pattern matching has more practical applications in computational biology, such as sequence alignments, motif finding, read mapping in gene and genome, substring matching, proteogenomics mapping, overlap detection, codon matching, etc. Thus, the problem is intentionally assumed here to search for the exact occurrences of the patterns (Kalsi, Peltola & Tarhio, 2008; Charalampos, Panagiotis & Konstantinos, 2011; Rivals, Salmela & Tarhio, 2011). There exists an impact of processing large sequences through the efficient algorithm, hence quantum algorithms are made suitable to process biological sequence applications. We search for multiple patterns set P={P1,.,Pk,.,Pm} with implicit consideration of processing singleton pattern set to find all t exact occurrence of single nucleotide patterns in gene and genome databases, or multiple nucleotide patterns to confirm the presence of amino acid within the peptide and protein sequences (Singh, 2015; Hakak & Kamsin, 2019). Later, in “proposed algorithmic applications to process biological sequences”, we define several applications of our quantum algorithms which are related to searching multiple patterns within the biological sequence databases. For a more comprehensive understanding to process the biological sequences, review these referenced articles (Sheik, Aggarwal & Anindya Poddar, 2004; Basel, 2006; Choo, 2006; Kalsi, Peltola & Tarhio, 2008; Fredriksson, 2009; Charalampos, Panagiotis & Konstantinos, 2011; Rivals, Salmela & Tarhio, 2011; Faro & Lecroq, 2013; Jiang, Zhang & Zhang, 2013; Singh, 2015; Zhang et al., 2015; Tahir, Sardaraz & Ikram, 2017; Hakak & Kamsin, 2019; Neamatollahi, 2020; Soni & Rasool, 2021; Soni & Malviya, 2021; Raja & Srinivasulu Reddy, 2019).

Motivation and contribution of work

The quantum machine can achieve computational speedups because of implicit parallelism. It needs O(1), i.e. constant execution step to realize an exponential number of operations (Nielsen & Chuang, 2010). We assume a problem of pattern matching as hard when the size of text database N=2n is excessively large as gigabytes (230), terabytes (240) or more (Kalsi, Peltola & Tarhio, 2008; Neamatollahi, 2020). So, instead of classical, the quantum pattern search takes reduced O(N=2n) time (Menon & Chattopadhyay, 2021). An existing quantum pattern matching solution achieved speedups over classical complexities (Ramesh & Vinay, 2003; De Jesus, Aborot & Adorna, 2013; Aborot, 2017; Soni & Rasool, 2021); however, the benchmark methods are constrained to find a single pattern, and the quantum multiple pattern matching is found ineffective because of executing multiple search oracles in successive iterations and it includes multiplicative factor m (Soni & Malviya, 2021).

The optimal quantum design may execute multiple search oracle in parallel on QPU with single-core to remove completely such factor m, however, this is impractical to design. We seek exact solutions of pattern matching with more applicability in computational biology. Thus, the available quantum benchmark algorithms QPBE and QBCE are enhanced here, with the names, enhanced QMEM processing-based exact algorithm (EnQPBEA-MPM) and enhanced quantum-based combined exact algorithm (EnQBCEA-MPM) for multiple pattern matching. The design of algorithms is based on processing effectiveness of QPU having C quantum cores and each core shares the text T on QMEM. So, to find all the t exact occurrences of each pattern, the search time complexities of the proposed algorithms are O((m/C)N) and O((m/C)t).

Our motivation is to search for all exact occurrences of m patterns either by direct use of effective quantum processing framework over original text sequence database T in O(N) queries or by transforming approximate filtering outcome into exactness over reduced search space in O(t) queries. The algorithms are based inherently on Grover’s search operator (GSO). We use QMEM to explore the text of size N=2n such that, the entire text search space is accessed in parallel in Ο(1) time, but memory word access needs O(log2N) steps (Park & Petruccione, 2019; Matteo, 2020; Soni et al., 2020). A new quantum circuit of Ο(1) time is proposed for exact match between pattern P and substring of T of size M, whereas classical comparison takes Ο(M) time (Sena Oliveira, Benicio Melo de Sousa & Viana Ramos, 2007; De Jesus, Aborot & Adorna, 2013; Soni & Rasool, 2021). Thus, we initiate quantum-effective algorithms with a context of exponential increase in biological text size. The proposed work of this article is organized as per Fig. 1, and we derive our results by giving the proofs of Theorem 1 and Theorem 2 in the proposed methods section.

Figure 1 Organization of quantum-based effective multiple pattern matching algorithms.

This article presents our main contribution as the effective quantum design of multiple patterns matching algorithms which are proved mathematically along with their simulations. We outline our work below to achieve objectives in a streamlined manner throughout this article: We realize the effective quantum processing framework by using QPU with C quantum cores which access quantum processing circuit of equivalent QMEM procedure. It achieves the quantum-based computational and processing speedups. We also proposed, a new constant time, quantum exact match (QEM) circuit which is utilized implicitly under GSO iterations.

We justify our proposed quantum algorithms using complexities analysis, and specific quantum proving techniques such as probabilistic, truthness and correctness proofs.

The future works of Soni & Rasool (2021) are presented here as enhanced solutions. Our proposed algorithms EnQPBEA-MPM and EnQBCEA-MPM are proved to search for all t exact occurrence of m patterns with effective time O((m/C)N) and O((m/C)t).

For a single pattern search, the proposed algorithms EnQPBEA-MPM and EnQBCEA-MPM can simulate the QMEM processing based enhanced designs of QPBE & QBCE algorithms (Soni & Rasool, 2021).

A factor (m/C) is proved negligible for a small arbitrary constant value of m and constant value of C as the QPU with C quantum cores utilizing their own set of quantum registers for searching m (m/C) is included explicitly in the time complexities for considering (m≫C) as the worst case.

The quantum operations of proposed algorithms are proved equivalent to their quantum circuits. These circuits are the actual realization of quantum solutions. Quantum query, time and storage complexities of proposed algorithms justify their effectiveness.

Based on several complexity analysis factors, we prove our proposed solutions as efficient to find exact patterns, and these remove the existing multiple pattern matching constraint (Soni & Malviya, 2021) as designs of QEMP and QAMP cannot exclude multiplicative factor m.

Our proposed quantum algorithms are simulated for validation through the quantum exact simulation toolkit (QuEST). Also, we proposed a quantum circuit implementation of QMEM through algebraic normal form (ANF). The intentions are not to analyze the efficiency of the simulation due to classical machine restrictions; therefore, we do the hybrid implementation.

We validate our results using QuEST simulation by assuming that t number of search solutions, either unique or multiple solution, are already known. To realize the case in which the value of t is unknown, we use quantum counting (QC) additionally to validate the search results of our proposed EnQPBEA-MPM and EnQBCEA-MPM algorithms.

We suggest several applications of EnQPBEA-MPM and EnQBCEA-MPM for processing biological sequences. Such applicability of these algorithms is specified with respect to significant characteristics and performance restrictions.

The abbreviated names used throughout the text are available in Table A1 (Appendix A). The nomenclature used in this article is a prerequisite for further reading purpose, therefore refer to Table B1 (Appendix B). The individual correctness proofs of algorithms EnQPBEA-MPM and EnQBCEA-MPM are separately included in Appendix C and Appendix D. A correctness proof shows algorithmic trace steps which expands the applied quantum operations.

Related Work

Prior work and the important findings

In classical findings, earlier exact multiple patterns matching solutions were proposed as the enhanced version of Knuth–Morris–Pratt (KMP) and Boyer–Moore (BM) algorithms. Both these multiple patterns matching solutions are available in Ο(mM) time for pattern pre-processing, and searching takes Ο(mN) (Zhang et al., 2015; Soni & Malviya, 2021). Based on these proposals, other existing algorithms are categorized for multiple pattern matching. There exist several multiple patterns matching methods, few are highlighted with their complexities. Aho–Corasick (AC) is the automata based prefix algorithm that works on KMP logic in O(m+N) time complexity. Commentz–Walter (CW) as a suffix algorithm, extending BM with possible variants, takes Ο(m(NM)) time in worst case. Multiple Pattern Backward DAWG (BDM) and Backward Set Oracle Matching (BSOM) are the factor or substring search-based algorithms which run in O(N(log|Σ|(mM)/M)) time (Charalampos, Panagiotis & Konstantinos, 2011; Faro & Lecroq, 2013). Instead, BSOM is a faster method; however, this needs extra space complexity of O(mM|Σ|) and verification through the AC algorithm. Wu–Manber (WM) is hashing based algorithm works for a large number of patterns search in O(N(⌈M/w⌉)) time, where w is number of bits in word size. Shift-OR (SO), Shift-AND (SA), and Backward Non-Deterministic DAWG (BNDM) Matching methods perform bits operation through intrinsic parallelism to realize solutions for multiple patterns matching in O(m(Nlog|Σ|(w)/w)) average time complexity (Fredriksson, 2009; Hendrian et al., 2019). The performance of these algorithms is dependent on |Σ|, size of text database |T|=N, number of patterns |P| = m and each pattern Pk with varying length Mk=[0 to Mk−1]. We noted that the multiplicative factor m is somehow included in time complexities of the classical algorithms. Among all the algorithms, AC has significant applications in biological sequence processing. However, AC requires large memory to store the automata, and hence it is constrained to process large patterns set. This algorithm induces competitive results on the small-sized |Σ| and |P|=m with each pattern Pk of short length Mk (Fredriksson, 2009; Charalampos, Panagiotis & Konstantinos, 2011; Faro & Lecroq, 2013; Zhang et al., 2015; Hendrian et al., 2019; Soni & Malviya, 2021).

There exist few solutions for quantum pattern matching with the advantage of using amplitude amplification of Grover’s search (GSO). This method finds search results over N-sized text in O(N) steps with high probability, and it is better than the classical linear search time Ο(N) (Lanzogorta & Uhlmann, 2008; Zhou et al., 2013; Coles, 2020). Few single and multiple patterns matching schemes are available in the quantum. Single pattern matching was initiated by Ramesh–Vinay (RV) through the quantum deterministic sampling method; however, the suggested solution needs O(M+N) time by including the pre-processing and searching (Ramesh & Vinay, 2003; Montanaro, 2017; Menon & Chattopadhyay, 2021). The method quantum approximate pattern matching (QAPM) filters the text for searching a pattern over reduced indices, and it needs O(t+t) (Aborot, 2017). A basic solution of quantum exact pattern matching (QEPM) finds exact leftmost occurrence of the pattern in O(N) time (De Jesus, Aborot & Adorna, 2013). The t time search is relatively better than N solution; however, QAPM finds approximate pattern match, and QEPM is constrained to search single pattern occurrence (De Jesus, Aborot & Adorna, 2013; Aborot, 2017).

Recent advancements of these algorithms are presented by extending the logic of QEPM or combining the methods of QEPM and QAPM for effective exact matching design. A suggested QMEM processing based exact (QPBE) algorithm is efficient to process large text sequences and this also overcomes the constraint of QEPM method by finding all t exact occurrences of search pattern in O(Nt) time (Soni & Rasool, 2021). However, the quantum-based combined exact (QBCE) algorithm replaces approximations of the QAPM method with exact matches. This also reduces implicit quantum circuit depth to explore the text during pattern search with logarithmic factors. The desired search time of all t exact occurrence of search pattern is O(t) (Soni & Rasool, 2021). Both these extended solutions are remarkable; however, no attempt has been made yet to design QPBE and QBCE algorithms to process multiple patterns, which are highly expected in biological sequence processing. As well as, the design of QBCE is not available under the specific processing of QMEM (Soni & Rasool, 2021). For quantum multiple pattern matching, initial solutions were suggested as an extension to QEPM and QAPM methods. Soni & Malviya (2021) suggested multiple pattern algorithms, renamed here as quantum exact multiple pattern (QEMP) algorithm to search for either the single occurrence or all t occurrence of m patterns within time complexities range O(mN) and O(mNt). Rather, renamed quantum approximate multiple pattern (QAMP) search for solution with suggested O(m(t+t)) time. Such algorithms search for the equal and unequal sized patterns in successive iterations, and this includes multiplicative factor m in time complexities. Thus, search solution for multiple pattern algorithms is not effective (Soni & Malviya, 2021).

We are providing the brief description of algorithms QEPM & QAPM, QPBE & QBCE, and QEMP & QAMP from the next subsection onward. As per the reviewed analysis of algorithms, the qubits estimations and algorithmic complexities analysis are also included separately.

Quantum exact pattern matching (QEPM) and quantum approximate pattern matching (QAPM) algorithms

The QEPM method is based on the design of an oracle that performs parallel matching under the quantum superposition by aligning comparison window between pattern and text substring to search for a leftmost exact occurrence of pattern. In O(N) queries, oracle inverts the leftmost index to report pattern match solution with high probability. The algorithm uses GSO and assures O(Nlog2N) search time complexity. Table 1 shows that the method is constrained to find a single (t=1) occurrence of pattern, and qubits estimation is the same as quantum search algorithm (De Jesus, Aborot & Adorna, 2013). The algorithm QAPM applies hamming distance (HD) method for approximate text filtering and matching. It cannot find accurate results as HD is an error model and allow replacement at unit cost. A pattern is reported when HD≤Threshold (pre-computed). A QAPM needs more storage as it uses a large number of quantum registers with excessive qubits requirement (Aborot, 2017); see Table 1. However, this searches all occurrences with O(t) queries. Additional indices may filter using HD, and this increases the size of filtered text, and even HD based verification generates search results with approximation. A filtering needs O(tlog2N) time to find all t filtered indices, then verification matches the t′ approximate occurrences of pattern through registers comparison, and t calls of GSO, in the O(tt′log2N) and O(tlog2N) time (Aborot, 2017).

Table 1 Analysis of algorithmic complexities QEPM and QAPM algorithms with qubits estimation.

Quantum algorithm	Matching occurrence	Pre-processing complexity	Existing quantum pattern searching time complexity	Storage complexity (qubits estimation)	
Query	Best time	Worst time	Generalized	
QEPM	Single	No Filtering	O(N)	O(Nlog2N)	O(Nlog2N)	O(N)	O(n+Mlog2|Σ|)	
QAPM	All	O(tlog2N)	O(tt′)	O(tt′log2N)	O(tlog2N)	O(t)	O(n+Mlog2|Σ|+log|Σ|M+M)	

QMEM processing based exact (QPBE) and quantum-based combined exact (QBCE) pattern matching algorithms

The recently proposed algorithms, QPBE and QBCE, are efficient to process the large text sequences. These also overcome the constraints of QEPM and QAPM methods (Soni & Rasool, 2021). The QPBE design is realized efficiently under the superposition of text indices on a quantum memory, and it finds all t exact occurrences of search pattern in O(Nt) time. However, the QBCE algorithm replaces pattern matching approximations with exact matches. It also reduces implicit quantum circuit depth to explore the text during pattern search with the logarithmic factor. A search time for all t′ exact occurrences over the reduced text of size t is O(tt′). Both these methods were proposed with significant aspects to process the specific biological sequences. A query remains the same as existing methods, rather the best & worst time complexity of QPBE is O(Ntlog2N) & O(Nlog2N), and QBCE time complexity is ranging between O(tt′log2t) and O(tlog2t) time. The search oracle for an exact pattern match is found efficient. As we reviewed, the storage complexity of QPBE is same as the existing QEPM algorithm. The QBCE remarkably reduces the storage while comparing with the excessive qubits requirement of pattern verification used in the existing QAPM method. All the required complexity analysis detail of these algorithms are included in Table 2 for quick reference (Soni & Rasool, 2021).

Table 2 Analysis of algorithmic complexities QPBE and QBCE algorithms with qubits estimation.

Quantum algorithm	Matching occurrence	Pre-processing complexity	Existing quantum pattern searching time complexity	Storage complexity (qubits estimation)	
Query	Best time	Worst time	Generalized	
QPBE	All	No Filtering	O(N)	O(Ntlog2N)	O(Nlog2N)	O(N)	O(n+Mlog2|Σ|)	
QBCE	All	O(tlog2N)	O(tt′)	O(tt′log2t)	O(tlog2t)	O(t)	O(n+Mlog2|Σ|+log2t)	

Quantum exact multiple pattern (QEMP) and quantum approximate multiple pattern (QAMP) matching algorithms

There are varieties of solutions proposed for the first time on quantum multiple pattern matching. However, the design of such algorithms is based on the execution of search oracle in successive iterations. So for different pattern sizes, it is iteratively called for finding all t occurrence of each pattern Pk. Due to this direct possible design, Table 3 shows that a multiplicative factor (m) is included in the complexities of algorithms (Soni & Malviya, 2021). The authors categorized multiple patterns matching methods as exact and approximate with quantum memory processing based design. Search complexity of suggested QEMP algorithm to find t occurrence of each Pk ranges between O((m)(Ntlog2N)) and O((m)(Nlog2N)). In contrast, the QAMP method uses approximate filtering in O(m(tlog2N)) time to find t filtered indices for each Pk. Further, the iterative search time for each Pk to search all t′ occurrences of reduced text of size t, ranges between O(m(tt′log2N)) and O(m(tlog2N)). Both these methods are not increasing storage complexity because of iterative executions; and reasonably, the multiplicative factor m is included. Still, there exist high qubits requirement in the QAMP algorithm (see Table 3). The time of quantum memory O(Nlog2N) is considered explicitly due to no such physical availability. These algorithms were suggested to process biological sequences. For the detailed design and analysis of these methods, refer to Soni & Malviya (2021).

Table 3 Analysis of algorithmic complexities QEMP and QAMP algorithms with qubits estimation.

Quantum algorithm	Matching occurrence	Pre-processing complexity	Existing quantum pattern searching time complexity	Storage complexity (qubits estimation)	
Query	Best time	Worst time	Generalized	
QEMP	Single	NoFiltering	O(mN)	O(m(Nlog2N))	O(m(Nlog2N))	O(m(N))	O(n+Mlog2|Σ|)	
All	NoFiltering	O(mNt)	O(m(Ntlog2N))	O(m(Nlog2N))	O(m(N))	O(n+Mlog2|Σ|)	
QAMP	All	O(m(tlog2N))	O(mtt′)	O(m(tt′log2N))	O(m(tlog2N))	O(m(t))	O(n+Mlog2|Σ|+log|Σ|M+M)	

Quantum Algorithmic Framework

Quantum operational framework used in the algorithmic design

Quantum algorithms based on superposition can perform exponential operations in parallel. The quantum behavior realizes qubit presence as |0⟩ and |1⟩ at same time. A |ψ⟩ is a column vector, represents superposition, and ⟨ψ| is a row vector; usually, Bra-Ket notation ⟨ψ|ψ⟩ is inner product. An n qubits quantum register |qi⟩ (qi∈{0,1}n) spans the tensor product of 2n dimension as Hilbert space. So, the computational basis is formed as |ψn⟩=α0|0⟩+…+αi|i⟩+…+α2n−1|2n−1⟩ to realize superposition under n dimensional vector space with complex probability amplitudes. Quantum registers can entangle with each other. A measurement collapses superposition into classical states |i⟩ between |0⟩ to |2n−1⟩ with probability |αi|2 such that ∑i∈{0,1}2|αi|2=1. To visualize such n qubits superposition with the required dimensions, refer to these article for the Bloch sphere model (Choo, 2006; Lanzogorta & Uhlmann, 2008; Nielsen & Chuang, 2010; Coles, 2020; Soni & Rasool, 2020).

Qubits remain in a pure state (vectors), but a quantum gate operator transforms n qubits into 2n×2n sized mixed state (density matrix). The outer product of vectors is obtained as |ψ⟩⟨ψ| because quantum unitary gate U applies certain operations within superposition to transform the quantum state. A U† is a conjugate transpose of U that performs the reverse quantum operation, such that UU†=I holds. Quantum logic operations such as [H] creates superposition, Pauli matrices [X,Y,Z] obtains any rotation on Bloch sphere, [Rx(θ),Ry(θ),Rz(θ)] applies rotation with angle θ as unitary operation. Some required controlled operations are [CnNOT or CnX] which flips the target qubit and [CnZ] flips the phase of target, when n control qubits are set to 1. The unitary operators encode to perform specific operations under quantum superposition. Refer to Table B1 (Appendix B) for symbols and used unitary operators throughout this article, and for more comprehensive understanding of quantum operations, refer to the following articles (Lanzogorta & Uhlmann, 2008; Nielsen & Chuang, 2010; Coles, 2020; Soni & Rasool, 2021; Soni & Malviya, 2021).

Methodology and framework used for quantum algorithm analysis

Our analysis framework for the quantum algorithm is oriented toward quantum-based proof methods. So, we categorized the proofs with their specialized point of interest with their additive use in the quantum algorithmic analysis. We provide the precise description of them as: Quantum Complexity Proof: The proposed algorithms are justified by using the following complexities analysis. Query complexity shows the number of superposition based oracle calls. Time complexity states the processing time of quantum gates involved in quantum circuits with logarithmic factors. Circuit complexity defines the composition of the quantum circuit with depth. Storage complexity estimates required qubits with ancilla.

Quantum Probabilistic Proof: The proposed algorithms are also proved based on computational theory to identify the quantum complexity class as either the exact quantum polynomial (EQP) with Pr=1 or bounded error quantum polynomial (BQP) complexity with Pr = ∈. The probabilistic proof is used to identify results based on probabilities to be used later in lemmas and theorems.

Quantum Truthness Proof: We prove the algorithms mathematically using Lemma proofs to derive primarily partial results, and then used Theorem proofs to justify the computational complexities result based on rigorous logic and reasoning.

Quantum Correctness Proof: We proved the proposed algorithms for their correctness on the basis of quantum algorithmic trace steps which expands quantum operations applied under superposition to show quantum state transformations.

(The above-mentioned proofs are categorized on the basis of the following references Lanzogorta & Uhlmann, 2008; Nielsen & Chuang, 2010; Faro & Lecroq, 2013; Zhou et al., 2015; Broda, 2016; Giri & Korepin, 2017; Grassi, Plasencia & Schrottenloher, 2018; Coles, 2020; Soni & Rasool, 2021; Soni & Malviya, 2021).

Quantum effective processing framework for algorithm design

We generalized a framework for the ordered design of algorithms with (1) processing advantage of quantum memory; (2) proposed efficient quantum exact match (QEM) circuit; and (3) Grover’s search to generate high probable results. The remarks of framework are specified in Table 4.

Table 4 Analysis of quantum memory processing, quantum exact match, and quantum search operation.

Quantum design	Quantum algorithmic requirement	Quantum unitary	Quantum time complexity	
Circuit (gates required)	Storage (qubits required)	Query	Best time	Worst time	
QMEM	Cn+1NOT:1, CNOT: Mlog2|Σ|	O(n+w)	UQMEM, USwap, ULoad	O(1)	O(log2N)	O(log2N)	
QEM	C2NOT: 3Mlog2|Σ|
Clog2|Σ|+1NOT: M, CM+1NOT: 1	O(M×log2|Σ|)	UComp	O(1)	O(1)	O(1)	
GSO	H: n+2n+1, X: 1+n+n
Cn+1NOT: 1, CnZ: 2	O(n+M×log2|Σ|)	UMark, UDiff	O(N)	O(Nlog2N)	O(Nlog2N)	

Processing advantage of quantum memory (QMEM)

First, for the compatibility of QPU based computation, both text and pattern are required to encode as quantum data. This facilitates the processing of large biological sequences under quantum superposition. So, QMEM of size |T2n×w⟩QMEM is used to realize superposition with N=2n memory words each with size w qubits. The QMEM needs address register |Ti⟩QA of size log2N=n qubits to refer all text indices |Ti⟩QA in superposition, and data corresponding to entangled addresses is accessed by data register |T[i]⟩QD of size log2|Σ|=w qubits (Giovannetti, Lloyd & Maccone, 2008; Nielsen & Chuang, 2010; Metodi, 2011; Lin et al., 2013; Fu et al., 2016; Britt, 2017).

The design of QMEM is realized using bucket brigade architecture that enables data access in O(log2N) steps, as among O(N=2n) qutrit, O(log22n) quantum switch remains active. This design is effective, as classical memory (CRAM) needs all O(N=2n) switches active for word access. So, QMEM gains exponential speedup as (2n/log22n) over CRAM. A text is shared on memory, and each cth core QCorec can access it in owned superposition. A QPU with C cores uses their registers set, and such parallelism minimizes processing time as negligible. Figure 2 shows the architecture of QPU with C cores working on shared QMEM with the design of the quantum memory circuit. A QMEM is realized using UQMEM of Eq. (1) in support of Eqs. (2)–(4) (Giovannetti, Lloyd & Maccone, 2008; Nielsen & Chuang, 2010; Metodi, 2011; Lin et al., 2013; Fu et al., 2016; Britt, 2017; Brandl, 2017; Park & Petruccione, 2019; Matteo, 2020; Soni et al., 2020).

(1) QMEM Transformation←(UQMEM←(U†Swap(ULoad(USwap))))

(2) USwap(|0⟩|wait⟩)=|f⟩|left⟩ or USwap(|1⟩|wait⟩)=|f⟩|right⟩

(3) ULoad(|Ti⟩QA⊗|T[w]⟩QD)=ULoad(|Ti⟩QA⊗|T[w⊕i]⟩QD)=|Ti⟩QA⊗|T[i]⟩QD

(4) U†Swap(|f⟩|left⟩)=|0⟩|wait⟩ or U†Swap(|f⟩|right⟩)=|1⟩|wait⟩

Figure 2 Quantum circuit equivalent to the quantum memory (QMEM) processing.

A unitary UQMEM makes the data available in parallel for each |Ti⟩QA index. It prepares the |qutritsN−1⟩ switches in |waitN−1⟩ state and realizes the superposition of entire memory. As per target index |Ti⟩QA, the qutrit are transformed from |wait⟩ to |left⟩ or |right⟩ state using USwap of Eq. (2) under fiduciary qubit |f⟩ that fixes switch state. Among |qutritsN−1⟩ only the |qutritslog2N⟩ remains active during the memory call [14]. We perform the data loading using ULoad of Eq. (3). It activates bus qubits to trace a path of active qutrit switches, copies the cell data, and traces back over same qutrit to load copied data into |T[i]⟩QD, and meanwhile, qutrit are transformed to |wait⟩ state by reverse unitary U†Swap of Eq. (4) (Giovannetti, Lloyd & Maccone, 2008; Nielsen & Chuang, 2010). A QMEM needs O(log2N) steps; and a memory call enables the bus qubits equal to the word size to access data in parallel, so for w=M×log2|Σ| qubits, the log2N switch remains active until the word transfer is not completed. Therefore, with word transfer, the QMEM needs O(M×log2N) steps with the negligible factor M (Nielsen & Chuang, 2010; Soni et al., 2020; Soni & Rasool, 2021; Soni & Malviya, 2021).

Proposed efficient quantum exact match (QEM) circuit

Second, we propose quantum-exact match (QEM) circuit through unitary UComp to perform parallel match between pattern |P[0 to M−1]⟩DR and retrieved substring in register |T[w]⟩QD of size w=M×log2|Σ| qubits. We seek an exact match on behalf of each index |Ti⟩QA in superposition on QMEM. This circuit compares the qubits of size log2|Σ| for each symbol contained in |P⟩DR. So, for M length pattern, all log2|Σ| sized qubits are analyzed in O(1) time. We specified the QEM operation in Eq. (5), and relevant circuit is shown in Fig. 3 with depth 2 i.e. O(1) (Sena Oliveira, Benicio Melo de Sousa & Viana Ramos, 2007; De Jesus, Aborot & Adorna, 2013; Soni & Rasool, 2021; Soni & Malviya, 2021).

(5) QEM Operation←(UComp:f(|Ti⟩QA)={0,if|T[i to i+M−1]⟩QD≠|P[0 to M−1]⟩DR1,if|T[i to i+M−1]⟩QD=|P[0 to M−1]⟩DR)

Figure 3 Quantum circuit for exact pattern match as (QEM) working with QMEM processing.

The comparison between |P[0 to M−1]⟩DR and |T[i to i+M−1]⟩QD is performed by unitary UComp in an explored superposition of QMEM with text indices |Ti⟩QA. So, entire w=M×log2|Σ| qubits sized substring is compared in parallel with constant time. This circuit is designed with 3×M×log2|Σ|=C2NOT gates which are arranged at level zero (for M sized text substring and pattern, and M additional ancilla). At level one, we used M=Clog2|Σ|+1NOT gates to check for equality as either |0⟩==|0⟩ or |1⟩==|1⟩ between aligned qubits of size log2|Σ| for each character of P. Last level is designed with single CM+1NOT gate that flips a target qubit |qComp⟩ to indicate the quantum-based exact match. So, the depth of quantum circuit is O(1). The qubits requirement of the proposed circuit is (3×M×log2|Σ|) + M + 1 and we estimate asymptotic complexity with O(M×log2|Σ|). Thus, this quantum-exact match (QEM) circuit is efficient.

Grover’s search operator (GSO) to generate high probable results

Third, Grover’s method is optimal to search for pattern in O(N) steps over N size text. It uses amplitude amplification that repeats for π/4N times, each iteration applies reflection operations for transforming target index to high amplified amplitude under superposition state, and thus to obtain high probable search results (Nielsen & Chuang, 2010; Chakrabarty, Khan & Singh, 2017). So, O(N) steps assure to eventually result in the desired state with significantly large amplitude. A method is shown in Fig. 4 and it’s next to next figure. No more iterations than N is recommended, as this succeeds with a solution on the sine function principle. It gradually increases as per the increase in function argument, but later this starts decreasing. However, this search mechanism is the only way to achieve a quadratic speedup (Lanzogorta & Uhlmann, 2008; Zhou et al., 2013; Coles, 2020).

Figure 4 Quantum-based illustration of the EnQPBEA-MPM algorithm.

The GSO operation is defined as Eq. (6) with sub-unitary specified in Eqs. (7) and (8). A search oracle O←(UMark(UComp)) marks the target index location of the pattern when UComp of Eq. (5) is succeeded for the exact match through Boolean oracle. Further, phase inversion is applied by phase oracle using UMark(|Ti⟩QA⊗|q⟩) of Eq. (7) to reflect the target index amplitude as |Ti⟩QA|−⟩ → (−1)f(|Ti⟩QA) |Ti⟩QA |−⟩. Another reflection operator diffusion D←(UDiff(O)), defined in Eq. (8), inverts all amplitudes around the mean, such that the amplitude of solution increases and the others decrease. In actual, this method amplifies the search index amplitude in each iteration (Lanzogorta & Uhlmann, 2008; Zhou et al., 2013; Broda, 2016; Giri & Korepin, 2017; Figgatt et al., 2017; Coles, 2020). The GSO operational description is provided in correctness proof of proposed algorithms.

(6) GSO Opeartion←(D←UDiff(O←(UMark(UComp))))

(7) O←UMark(|Ti⟩QA⊗|q⟩)=(I−2|Ti⟩QA⟨Ti|QA)⊗(|Ti⟩QA⊗|q⟩)={|Ti⟩QA,f(|Ti⟩QA)=0−|Ti⟩QA,f(|Ti⟩QA)=1

(8) D←UDiff(O)=D((2|ψn⟩⟨ψn|−I)(O))=D(H⊗n(2|0⟩⟨0|−I)H⊗n(O))

A GSO works under superposition state |ψn⟩ by making an angle (π/2−θ) and transforms the solution state by applying each time 2θ rotations. So, for r rotations, (r×2θ=π/2−θ) then (r=π/4θ−1/2). In superposition of N elements, the amplitude for each of t solutions are t/N, so θ=t/N. On UComp success Pr[GSO outputs|Ti⟩QAif f(|Ti⟩QA)==1]=t/N. So, put this phase θ in r we get (r=π/4N/t−1/2)≈π/4N/t≅O(N/t) step. For geometric proof, refer (Nielsen & Chuang, 2010; Broda, 2016; Chakrabarty, Khan & Singh, 2017; Coles, 2020; Soni & Rasool, 2021; Soni & Malviya, 2021). The 1≤t≤N solutions are searched in N/t query and O(N/tlog2N) time with Pr[GSO outputs|Ti⟩QAif f(|Ti⟩QA)==1]≥t/N.

As reviewed, analysis of quantum effective processing framework is specified in Table 4. The QMEM makes searching outcomes available in parallel with O(1) step. The unitary UComp of O(1) time is used in GSO as implicit operation and it is simulated on QMEM design. The GSO can find number of pattern occurrences t=tfew (multiple solution), where tfew denotes few pattern occurrences ( tfew≪N), using O(N/t) queries and O(N/tlog2N) time. As H⊗n operations run in parallel, each with O(1) time, so asymptotic complexity is considered as O(N/t) with respect to negligible multiplicative factor (log2N) (Brassard et al., 2002; Lomont, 2003; Ablayev et al., 2020). If it is known that the t=0 (no solution), then GSO returns a random element uniformly in O(N) time. In case, t=1 (unique solution), search result can be obtained in O(N) time with high probability.

To report t search solution, GSO needs t×N/t=t×t×N/t=Nt queries and O(Ntlog2N) time. However, all solutions t=1 or tfew are possible in O(N) and when t=N (all are search solution), the search time is O(N) i.e. same as the classical. For t=1 case, GSO can obtain the result with high probability after N iterations, however, more N iterations can again generate uniform probability. This may happen repeatedly in each successive N iterations. Therefore, only O(N) iterations are needed to obtain high probable search solution. We prefer the quantum search to find the few pattern occurrences. The consideration of t=N is found rare for a biological text, and hence this can be ignored (Nielsen & Chuang, 2010; Broda, 2016; Chakrabarty, Khan & Singh, 2017; Soni & Rasool, 2021; Soni & Malviya, 2021).

To our knowledge, the quantum search assumes that the number of search solutions t (either unique or multiple solution) are already known. Therefore, number of GSO iterations can be determined in advance and after π/4(N/t) iterations the search results are found with certainty and high probability. However, the GSO can overshoot if the t number of search solutions are unknown/not known in advance. In that case, with the unknown number of GSO iterations, the probability of success would be vanishingly small (Boyer et al., 1998; Brassard et al., 2002; Lomont, 2003; Younes, 2008; Song, 2017; Ablayev et al., 2020).

To deal with the unknown number of search solutions, one of the methods was proposed by Boyer et al. (1998) and restated in Younes (2008); Song (2017) and Ablayev et al. (2020) as the modified Grover’s search that runs GSO several times in successive iterations. The modified algorithm of Boyer et al. (1998) repeats GSO by taking the value of t in an exponential increase. On jth repetition, π/4(N/2j) iterations are performed. The repetitions are here summing to O(N) times. Either of these iterations may find the search results with a sufficient high probability. In each of these repetition, the GSO operations are still bounded by π/4(N) iterations. It is equivalent to O(N) time classical complexity, so not used in practical implementation (Boyer et al., 1998).

Quantum counting (QC) is an alternative approach that can satisfactorily handle the problem of unknown number of search solutions (Brassard et al., 2002; Lomont, 2003; Song, 2017). A QC is quantum amplitude estimation (QAE) method that can estimate t number of search solutions either based on approximation or based on exactness. It helps to decide the required number of GSO iterations. The QAE technique is defined in Brassard et al. (2002) and Fang Song (2017), and it is used for estimating t=|{|Ti⟩QA∈N|f(|Ti⟩QA)==1}| as the possible count to find the number of search solutions. These authors (Boyer et al., 1998; Brassard et al., 2002; Lomont, 2003) suggested to run quantum counting algorithm initially, and then to proceed with actual number of GSO iterations. Quantum counting results can be obtained with quadratic speedup in O(N) time. Therefore, we observed it as an efficient method when the number of search solutions are unknown, and hence it prevents overshooting of Grover’s. Later, in theoretical results and complexity analysis section, we analyze the exact and approximate quantum counting methods, and these are implemented to simulate our algorithms.

A circuit of QMEM needs Cn+1NOT to mark |Ti⟩QA address, and to store w=M×log2|Σ| in |T[w]⟩QD we use M×log2|Σ|=CNOT. This memory is exponentially faster than the CRAM circuit. However, its access depends on the depth of the bifurcation tree i.e. O(log2N) time. Quantum search works with QMEM by applying H⊗n(|Ti⟩QA) and XH(|q⟩) and then UComp checks for exact match followed by amplification. On a successful match, qubit |q⟩ is flipped by Cn+1NOT gate, then UMark flips the phase of index |Ti⟩QA by CnZ gate. Diffusion performs the amplification through the set of quantum operators {{H⊗nX⊗n}CnZ{H⊗nX⊗n}}. At the last, perform measurement at index |Ti⟩QA. In addition, we included qubits requirement for QMEM and GSO. Quantum gates and the circuit requirement of the framework is shown in Table 4. However, our remark states that quantum search over text T of size N takes n+2Mlog2|Σ|+1 qubits (Lanzogorta & Uhlmann, 2008; Nielsen & Chuang, 2010; Zhou et al., 2013; Broda, 2016; Chakrabarty, Khan & Singh, 2017; Giri & Korepin, 2017; Figgatt et al., 2017; Coles, 2020; Soni & Rasool, 2021; Soni & Malviya, 2021). Further, n is replaced by tq qubits for the search which is performed over the reduced size filtered text. A QMEM is efficiently simulated using algebraic normal form (ANF) for the hybrid realization of quantum operations (Bogdanova et al., 2018; Malviya & Tiwari, 2020; Hao et al., 2020; Malviya & Tiwari, 2021).

The Proposed Methods

This section includes proposed EnQPBEA & EnQBCEA algorithms. Both these designs use the effective quantum processing framework. Algorithms can process multiple patterns string of set P={P1,.,Pk,.,Pm} with each pattern Pk of length Mk (1≤k≤m), using the shared text T of size N explored on QMEM to search all exact match occurrence of individual pattern Pk∈P through cth core QCorec of QPU having C quantum cores. Our algorithms are enhancement of improved QPBE & QBCE methods for processing multiple patterns with an aim to remove the multiplicative factor m in complexities. The proposed solutions are remarkable and efficient on comparing with existing QEMP & QAMP multiple pattern methods. We modify the design of algorithms by running multiple search oracles in parallel. A QPU runs C cores to search for m/C pattern in parallel, and each quantum core uses its own set of registers. So a multiplicative constant (m/C) with a small arbitrary constant value of m and constant value of C is found negligible. However, for comparatively large value of m≫C, a factor m/C cannot be ignored in the complexities analysis. Hence, we initially clarify that for few pattern occurrences, the storage and time both are implicitly saved in enhanced designs of algorithms. We justify our proposed methods by giving the proof of the resulting Theorems 1 and 2. Later, we show the efficient and effective hybrid simulation of these quantum algorithms.

Proposed method 1: enhanced QMEM processing based exact algorithm for multiple pattern matching (EnQPBEA-MPM)

This method searches for each pattern Pk∈P in parallel using QPU with C cores accessing text T on shared QMEM, such that search time of all tk occurrence of Pk overlaps. QEM circuit is applied under superposition of text on QMEM by each QCorec. Search results are instantly possible and would be effective for the biological sequencing because of no other processing overhead except the search time. Existing QPBE is enhanced efficiently by executing search oracles in parallel with the negligible time factor, and the existing iterative pattern search overhead of QEMP is also removed.

A pattern Pk∈P is individually processed on cth core QCorec where (1≤c≤C), so m/C patterns are searching in parallel within the text T of size N shared on QMEM. Each pattern Pk is assumed with individual size w=Mk∗log2|Σ| qubits, and it is stored in |P[0 to Mk−1]⟩DRk as in separate data register. The text T realized on |T2n×w⟩QMEM is accessed in a superposition of addresses by QCorec through address register |Ti⟩QAk (i∈{0,1}n). All the text substrings, each of length Mk∗log2|Σ| are loaded in entangled register |T[w]⟩QDk by applying QMEM transformation. A unitary ULoad makes sure such data load in a coherent superposition of text addresses. Once these substrings are available in parallel, EnQPBEA applies the GSO operator, separately on QCorec to ensure an exact match of each Pk with QEM circuit realized using UkComp. The Boolean oracle circuit succeeds by flipping target qubit of Fig. 3 to report exactness. When cth core QCorec identifies exact match in superposition, the amplification operator UDiff(UMark) is then applied to increase the probability amplitude of identified indices |Ti⟩QAk. The GSO operator repeats for O(N) time and then QCorec applies the measurement to obtain search index |Ti⟩QAk with high probability.

Proposed Algorithm 1: EnQPBEA-MPM.

	Data	:	Text T stored on |T2n×w⟩QMEM which is accessed by quantum registers {|Tn⟩QA1,…,|Tn⟩QAm} and {|T[w]⟩QD1,…,|T[w]⟩QDm}, the implicit data registers {|P[0 to M1−1]⟩DR1,…,|P[0 to Mm−1]⟩DRm} each of size w=Mk×log2|Σ| to store search pattern Pk∈P={P1,.,Pk,.,Pm}, and set of ancillary qubit designated to number of patterns {|q⟩Q1,…,|q⟩Qm}	
	Result	:	Outputs all tk exact occurrence (1≤tk≤N) of each pattern Pk∈P in parallel using cth quantum core QCorec accessing T on shared QMEM, as index |Ti⟩QAk s. t. |T[i to i+Mk−1]〉QDk==|P[0 to Mk−1]〉DRk	
1:	Procedure EnQPBEA-MPM	
2:		Prepare registers as |zeroesn⟩ in |Tn⟩QAk, |zeroes[w]⟩ in |T[w]⟩QDk, |1⟩ in |q⟩Qk and |P[0 to Mk−1]⟩DRk	
3:	For each pattern Pk∈P to be processed separately on cth quantum core QCorec	
4:		Initialize quantum state in registers as |ψn⟩k in |Ti⟩QAk, |same[w]⟩ in |T[w]⟩QDk & |q⟩Qk as |−⟩	
5:	For all |Ti⟩QAk in their separate uniform quantum superposition state |ψn⟩k	
6:		Load data at |T[i]⟩QDk as per entangled |Ti⟩QAk by applying QMEM Transformation as	
7:		ULoad(|Ti⟩QAk⊗|T[w]⟩QDk)=ULoad(|Ti⟩QAk⊗|T[w⊕i]⟩QDk)=|Ti⟩QAk⊗|T[i]⟩QDk	
8:	Repeat GSO for O(N/tk) times in uniform superposition |ψn⟩k, with QEM Operation which is implicitly applied through UkComp for exact matching of Mk×log2|Σ| qubits size as –	
9:		UkComp: f(|Ti⟩QAk)={0,if|T[i to i+Mk−1]⟩QDk≠|Pk[0 to Mk−1]⟩DRk1,if|T[i to i+Mk−1]⟩QDk=|Pk[0 to Mk−1]⟩DRk	
	GSO Opeartion←(D←UDiff(O←(UMark(UComp))))	
10:	End of GSO Repeat	
11:	Measure the final state to get the desired index |Ti⟩QAk as high probable solution	
12:	Verify pattern Pk at |Ti⟩QAk on cth core QCorec as |T[i to i+Mk−1]⟩QDk==|P[0 to Mk−1]⟩DRk	
13:	End of Inner For	
14:	End of Outer For	
15:	End Procedure	

The quantum state gets collapsed after each measurement, so its repetition ensures to report all tk index locations of Pk∈P which are identified by cth quantum core QCorec. Each pattern occurrence is verified by same core as |T[i to i+Mk−1]⟩QDk==|P[0 to Mk−1]⟩DRk. The pattern matching method of EnQPBEA-MPM is illustrated in Fig. 4. However, the steps are listed in the proposed algorithm, and the equivalent quantum circuit executing search oracles in parallel is shown in Fig. 5. In reference to Table 4 discussion, we state, that each core realizes O(N/tk) iterations of GSO in parallel, and therefore, results in all desired pattern occurrence tk on behalf of pattern Pk. However, we require O(Ntk) queries to report all tk marked occurrences, and hence the pattern matching time is bounded to O(Ntklog2N) with negligible logarithmic factor. We clarify that EnQPBEA repeats GSO operation in parallel for tk times on each core QCorec to search all tk indexes. So, we consider t=tk=max(t1,.,tk,.,tm) as based on longest core processing to find maximum pattern occurrences. Therefore, the search complexity of parallel executions of EnQPBEA using QPU with C cores is O((m/C)(Ntlog2N)) time. In support of complexities, a correctness proof of EnQPBEA-MPM with quantum operations is specified in Appendix C. For mathematical proof, we define certain Lemma 1 as partial required proof, and based on that, we conclude the computational complexity and achieved speedup through the resulting proof of Theorem 1.

Figure 5 Quantum circuit equivalent to search mechanism of EnQPBEA-MPM algorithm.

Lemma 1: A QPU having C quantum cores (1≤c≤C) can access the text T of size N on shared QMEM. It loads all text substring equal to pattern length Pk as Mk∗log2|Σ| qubits in superposition by using QCorec in parallel. Time needed for such parallel loading operations ranges between O((m/C)(Mlog2N)) and O((m/C)(MNlog2N)).

Proof (Lemma 1): About earlier discussions of effective processing framework and Table 4, we use to prove this lemma. The |T2n×w⟩QMEM as shared among C cores of QPU, makes sure that all |Ti⟩QAk addresses are available in parallel on each QCorec & QMEM transformation loads Mk∗log2|Σ| qubits in entangled register |T[w]⟩QDk. The entire memory access is available in constant time on each individual core, however, by considering M as M=max(M1,.,Mk,.,Mm) the memory circuit needs O(Mlog2N) steps. So, the parallel time of QMEM access using QPU with C quantum cores is O((m/C)(Mlog2N)). As we know, that the quantum state is collapsed after each measurement, so to report all tk index of Pk identified by cth quantum core QCorec, we need this access several times. By assuming t=tk=max(t1,.,tk,.,tm) at any QCorec for QMEM transformation, and at worst, if number of identified patterns are t=N then each time, all parallel substring load will take O((m/C)(MNlog2N)) time. We discussed earlier, that both these factors M and (m/C) are negligible due to parallel load and parallel processing by achieving exponential speedup.

Theorem 1: Given text database T of size N and the multiple patterns set P={P1,.,Pk,.,Pm} with each pattern Pk of length Mk(1≤k≤m). Algorithm EnQPBEA-MPM uses QPU having C quantum cores (1≤c≤C) to access the text T on shared QMEM. A cth core is used to search for the all tk exact occurrence of a pattern Pk indexed at |Ti⟩QAk, that is |T[i to i+Mk−1]⟩QDk==|P[0 to Mk−1]⟩DRk. Based on longest core processing to find pattern occurrences t=max(t1,.,tk,.,tm), the search time complexity of EnQPBEA-MPM is O((m/C)(Ntlog2N)) in the best case and O((m/C)(Nlog2N)) for the worst case.

Proof (Theorem 1): This proof relies on Lemma 1 and other statements which are justified earlier. Proof of Lemma 1 states that, for t=1 or tfew ( tfew denotes few pattern occurrences ( tfew≪N)) and t=N, all substring load transformation is possible in O(log2N) and O(Nlog2N) time. Now EnQPBEA-MPM algorithm realizes such parallelism using QPU with C quantum cores and each core access text T on shared QMEM. For each pattern Pk∈P of length Mk(1≤k≤m), this algorithm identifies the target indices based on the QEM circuit under superposition of N sized text. Further, the simultaneous iterations of GSO finds all tk solutions of Pk using cth quantum core QCorec in O((m/C)(N/tk)) queries.

Indeed, quantum state is collapsed while measured, so, EnQPBEA repeats GSO followed by measurement on QCorec to report all tk occurrence of Pk in O((m/C)(Ntk)) queries. Now, based on longest core processing, consider t=tk=max(t1,.,tk,.,tm). So, using QPU with C cores and for t=1 or tfew ( tfew denotes few pattern occurrences ( tfew≪N)), the best case time complexity of EnQPBEA-MPM is O((m/C)(Ntlog2N)) and this finds all patterns in parallel. However, when t=N (all are search solutions), the worst-case time complexity is O((m/C)(Nlog2N)). A multiplicative factor (log2N) is considered negligible with n qubits, surprisingly small, to expand the original search space. However, this factor cannot be ignored when the number of qubits n is usually large to expand the original text space (Lomont, 2003). And the multiplicative constant (m/C) with a small arbitrary constant value of m and constant value of C is found negligible. However, for the comparatively large value of m≫C, a factor m/C cannot be ignored in time complexities. Therefore, quantum search is preferred effectively for finding few occurrences. Instead, for biological text, t=N is rare and hence ignored while stating the generalized complexity. We know that algorithm design is based on GSO, so, results are obtained with at least probability as Pr[EnQPBEA running atQCorec measures |Ti⟩QAkin each iteration]≥tk/N.

Proposed method 2: enhanced quantum-based combined exact algorithm for multiple pattern matching EnQBCEA-MPM

The algorithm EnQBCEA-MPM is an enhanced version of the existing benchmark method QBCE. So, we formalize this multiple pattern algorithm with the possible speedup. The pattern matching method is illustrated in Fig. 6. Each pattern Pk∈P is individually processed using QPU with C cores; however, the cth core QCorec (1≤c≤C) processes the text T of size N over shared QMEM for (m/C) patterns either for filtering or searching. In this method, each core QCorec transforms the original N sized text into reduced search space tk (corresponding to Pk), so-called filtered indices, and then performs exact searching of all tk′ occurrence of each Pk in overlapping of time evolution. To transform the text into reduced search space, we use an existing method of quantum-approximate filtering (QAF). This method is based on the hamming distance (HD) to check for the possible errors between pattern and text substring (to filter index) and ensures its correctness when the hamming distance (HD)≤threshold (pre-computed). Such filtering outcomes are based on approximations, thus, we verify the filtered indices for a pattern match using the exactness. An additional time of QAF filtering is included in the complexity of this algorithm; however, this allows searching of patterns in an optimized way by achieving speedup. Our EnQBCEA design executes exact search oracles in parallel with the negligible time factor, and this also removes the existing iterative overhead (text filtering and pattern searching) of the QAMP algorithm. We expect the pattern matching results as effective for the biological sequencing because of overlapped quick search time to find the exact matches over the filtered text indices.

Figure 6 Quantum-based illustration of the EnQBCEA-MPM algorithm.

Initially, we redefine QAF (Aborot, 2017) to execute for each QCorec while accessing text T on shared QMEM for text filtering. The procedure QAF is redesigned here by using QMEM transformation. This prepares the pattern in register |P⟩DR and the start locations of distinct symbols of the pattern are store in array SL[M]. Now, initializes QMEM registers |Tn⟩QA and |T[w]⟩QD in the zero state along with auxiliary register |Tn+1⟩AX measured for the filtered index as possible start locations of the pattern. The superposition of text created in |Ti⟩QA and |Tn+1⟩AX is made entangled with addresses. Under memory superposition of |Ti⟩QA, QAF marks the distinct symbols of the pattern at |Tij⟩AX by unitary USLoc. And then, the possible start location of the patterns are marked as |Ti−Tij⟩AX by UPLoc. A Hamming distance (HD) is applied at |Ti−Tij⟩AX to check for threshold, further, Hadamard is applied at |Ti⟩QA to merge probability amplitudes of entangled indices of |Ti−Tij⟩AX. Finally, measure auxiliary register |Ti−Tij⟩AX to identify filtered indices |Ti⟩ which are then stored in the referenced location array LA[…]. As measurement destroys quantum state, so in each call at cth quantum core QCorec on behalf of Pk∈P, the QAF needs its execution several times to filter all tk indices location and then to store within the location array LAk[tk].

Procedure: QAF(|Tn⟩QA,|P⟩DR,LA[…]).

	Input	:	Text address register |Tn⟩QA, auxiliary register of same size with additional qubit |Tn+1⟩AX, the implicit data register |P[0 to M−1]⟩DR, SL[M] classical array that keeps distinct symbol location within pattern as |ij⟩, access to location array LA[…] to classically store filtered text indices.	
	Output	:	Stores all filtered text indices as possible start of pattern in location array LA[…]	
1:	Begin Procedure	
2:	Prepare P in |P[0 to M−1]⟩DR, and store |ij⟩ in SL[M] as preprocessed start locations of distinct symbol of |P⟩DR	
3:	Prepare registers as |zeroesn⟩ in |Tn⟩QA, |zeroes[w]⟩ in |T[w]⟩QD, |zeroesn+1⟩ in |Tn+1⟩AX	
4:	Initialize State as |ψn⟩ in |Ti⟩QA, |same[w]⟩ in |T[w]⟩QD & entangle the register |Tn+1⟩AX	
5:	Load data at |T[i]⟩QD as per entangled |Ti⟩QA by applying QMEM Transformation as	
6:		ULoad(|Ti⟩QA⊗|T[w]⟩QD)=ULoad(|Ti⟩QA⊗|T[w⊕i]⟩QD)=|Ti⟩QA⊗|T[i]⟩QD	
7:	For each |Ti⟩QA remains in uniform quantum superposition state |ψn⟩ do	
8:		Mark distinct symbol of pattern by unitary USLoc as |Tij⟩AX corresponding to |Ti⟩QA	
9:	Mark possible start location of pattern by UPLoc as |Ti−Tij⟩AX on behalf of |Ti⟩QA	
10:	Apply HD at |Ti−Tij⟩AX to check for distance between text and pattern, such that, HD≤ threshold	
11:	Apply Hadamard at |Ti⟩QA to merge amplitudes of entangled indices of |Ti−Tij⟩AX	
12:	Measure the auxiliary register |Ti−Tij⟩AX and store the identified index as |Ti⟩ in LA[…]	
13:	End of For	
14:	End Procedure	

Algorithm EnQBCEA-MPM needs following preparation such as – each pattern Pk∈P is assumed with individual size w=Mk∗log2|Σ| qubits, and it is stored in |P[0 to Mk−1]⟩DRk as in separate data register. At first, procedure QAF(|Tn⟩QAk,|Pk⟩DRk,LAk) is called for each pattern Pk on each core to store the filtering results at individual location array LAk[tk]. Each LAk[…] contained with tk≤N filtered text indices; therefore, the algorithm needs location register |Ttq⟩QLk each of size log2tk=tq qubits to access LAk[tk] by using cth core QCorec.

Proposed Algorithm 2: EnQBCEA-MPM.

	Data	:	Text T stored on |T2n×w⟩QMEM which is accessed by quantum registers {|Tn⟩QA1,…,|Tn⟩QAm} and {|T[w]⟩QD1,…,|T[w]⟩QDm}, the implicit data registers {|P[0 to M1−1]⟩DR1,…,|P[0 to Mm−1]⟩DRm} each of size w=Mk×log2|Σ| to store search pattern Pk∈P={P1,.,Pk,.,Pm}, separate location arrays {LA1[…],…,LAm[…]} to classically store {t1,..,tm} filtered text indices corresponding to each Pk, location registers to access filtered indices for each pattern as {|Ttq⟩QL1,…,|Ttq⟩QLm} each with size log2tk=tq qubits, & set of ancillary qubits designated to no. of pattern {|q⟩Q1,…,|q⟩Qm}	
	Result	:	Outputs all tk′ exact occurrence (1≤tk′≤tk) of each pattern Pk∈P in parallel, using cth quantum core QCorec accessing filtered location array LAk[tk] which is explored on QMEM, as searched index |TiLk⟩QAk s.t. |T[iLk to iLk + Mk−1]⟩QDk==|P[0 to Mk−1]⟩DRk	
1:	Procedure EnQBCEA-MPM	
2:		For each pattern Pk∈P to be processed separately on cth quantum core QCorec	
3:		Call QAF(|Tn⟩QAk,|Pk⟩DRk,LAk);	
4:	End of For	
5:	Prepare registers as |zeroestq⟩ in |Ttq⟩QLk, |zeroesn⟩ in |Tn⟩QAk, |zeroes[w]⟩ in |T[w]⟩QDk, |1⟩ in |q⟩Qk and |P[0 to Mk−1]⟩DRk	
6:	For each pattern Pk∈P to be processed separately on cth quantum core QCorec	
7:		Initialize quantum state for accessing LAk[tk] in register as |ψtq⟩k in |Ti⟩QLk, |T[i]⟩QLk in |Ti⟩QAk, |same[w]⟩ in |T[w]⟩QDk, & |q⟩Qk as |−⟩	
8:	For all |Ti⟩QLk in their separate uniform quantum superposition state |ψtq⟩k	
9:		Apply the unitary UkGetL to get n-qubits actual index as |T[i]⟩QLk=iLk i.e. the memory content of LAk[tk] through |Ti⟩QLk, and then store kth address in corresponding register |TiLk⟩QAk	
10:		UkGetL: f(|Ti⟩QLk)= |Ti⟩QLk|T|T[i]⟩QLk⟩QAk→ |Ti⟩QLk|TiLk⟩QAk	
11:	Load data at |T[iLk]⟩QDk as per addresses |TiLk⟩QAk by applying QMEM Transformation as	
12:		ULoad(|TiLk⟩QA⊗|T[w]⟩QD)=ULoad(|TiLk⟩QAk⊗|T[w⊕i]⟩QDk)=|TiLk⟩QAk⊗|T[iLk]⟩QDk	
13:	Repeat GSO for O(tk/tk′) times in uniform superposition |ψtq⟩k, with QEM Operation which is implicitly applied through UkComp for exact matching of Mk×log2|Σ| qubits size as –	
14:		UComp: f(|TiLk⟩QAk)={0,if|T[iLk to iLk+Mk−1]⟩QDk≠|P[0 to Mk−1]⟩DRk1,if|T[iLk to iLk+Mk−1]⟩QDk=|P[0 to Mk−1]⟩DRk	
15:	GSO Opeartion←(D←UDiff(O←(UMark(UComp))))	
16:	End of GSO Repeat	
17:	Measure the final state to get the desired index |TiLk⟩QAk as high probable solution	
18:	Verify pattern Pk at |TiLk⟩QAk on cth core QCorec as |T[iLk to iLk+Mk−1]⟩QDk==|P[0 to Mk−1]⟩DRk	
19:	End of Inner For	
20:	End of Outer For	
21:	End Procedure	

The algorithm EnQBCEA proceeds to search for each pattern by running all cores in parallel. An equivalent quantum circuit executing search oracles in parallel is shown in Fig. 7. So, each core QCorec explores filtered indices of LAk[tk] in superposition over QMEM. We expect that the reduced search space of size tk≪N is small than that of original text T of size N. Algorithm prepares registers in |zero⟩ states, and initializes superposition of filtered indices for each LAk[tk] as |ψtq⟩k by using |Ti⟩QLk(i∈{0,1}tq) of QCorec. Now, for each |Ti⟩QLk under the quantum superposition |ψtq⟩k we apply UkGetL to obtain n−qubits original filtered index as |T[i]⟩QLk=iLk i.e. [i]th memory content of LAk[tk] through |Ti⟩QLk. This transformation helps address register |TiLk⟩QAk to access actual indices, so the search can perform over original text.

Figure 7 Quantum circuit equivalent to searching logic of the EnQBCEA-MPM algorithm.

Now, the original text is available in QMEM superposition and shared among all quantum cores. Therefore, text T realized on |T2n×w⟩QMEM is accessed in a superposition of addresses by QCorec by address register |TiLk⟩QAk (iLk∈{0,1}n). All the text substrings, each of length Mk∗log2|Σ| are loaded in entangled register |T[iLk]⟩QDk by applying QMEM transformation. A unitary ULoad makes sure such loading is in a coherent superposition of text addresses. Once substrings are available in parallel, EnQBCEA applies GSO, by QCorec to find an exact match of each Pk with QEM circuit realized using the unitary operator UkComp.The processing of GSO is the same as per earlier discussion of EnQPBEA and Table 4. Instead, each core realizes O(tk/tk′) iterations of GSO in parallel, however, after O(tk) repetition QCorec applies measurement to obtain index |TiLk⟩QAk with high probability. As measurement collapses quantum state, so, each QCorec requires O(tktk′) queries to report all tk′≤tk marked occurrences of Pk in T. In addition, EnQBCEA-MPM allows cth core QCorec to verify pattern match at |TiLk⟩QAk as |T[iLk to iLk+Mk−1]⟩QDk==|P[0 to Mk−1]⟩DRk. A correctness proof of EnQBCEA-MPM is included in Appendix D and complexity is proved in Theorem 2.

Lemma 2: A QPU having C quantum cores (1≤c≤C) can access the text T of size N on shared QMEM. A cth core filters tk indices |Ti⟩QAk in parallel to identify the possible start locations of pattern P and to store such original filtered indices in LAk[tk]. Time needed for executing quantum approximate filtering (QAF) in parallel is O((m/C)(tlog2N)).

Proof (Lemma 2): In support of the earlier discussions and using the reference to Aborot (2017) and Soni & Rasool (2021), we used to prove this lemma. Procedure QAF is executed in parallel for each pattern Pk∈P on cth quantum core QCorec sharing the QMEM. Quantum circuit included in Aborot (2017) and Soni & Rasool (2021) will runs separately on QCorec and performs equivalent quantum operations as USLoc, UPLoc, HD followed by the Hadamard on |Ti⟩QAk to merge probability amplitudes of entangled indices of |Ti−Tij⟩AX. Now auxiliary register measures filtered indices |Ti⟩ to store in LA[…]. All such operations are bounded by O(log2N) time. However, measurement destroys quantum state, so in each call at cth quantum core QCorec on behalf of Pk∈P, the QAF needs its repeated executions to filter all tk indices and to store them in location array LAk[tk]. Therefore, based on longest core processing to filter maximum pattern locations, we assume t=tk=max(t1,.,tk,.,tm) at any QCorec. The time required for such filtering in parallel is O((m/C)(tlog2N)). The multiplicative factors can be ignored due to parallel processing – quantum circuit operations.

Theorem 2: Given text database T of size N and the multiple pattern set P={P1,.,Pk,.,Pm} with each pattern Pk of length Mk(1≤k≤m). Algorithm EnQBCEA-MPM uses QPU having C quantum cores (1≤c≤C) to access the text T on shared QMEM. The cth core runs QAF to store all tk filtered indices of a pattern Pk in LAk[tk]. The indices of LAk[tk] are used by cth core to search for all tk′ exact occurrence of patterns indexed at |TiLk⟩QAk, that is |T[iLk to iLk+Mk−1]⟩QDk==|P[0 to Mk−1]⟩DRk. Based on maximum filtered indices t=max(t1,.,tk,.,tm) in LAk[t] and longest core processing to find pattern occurrences t′=max(t1′,.,tk′,.,tm′), the search time complexity of EnQBCEA-MPM algorithm is O((m/C)(tt′log2t)) in the best case and O((m/C)(tlog2t)) for the worst case.

Proof (Theorem 2): The proof of this theorem is based on Lemma 2 and other statements which are justified earlier. Our algorithm EnQBCEA-MPM performs a search on filtering outcomes that are stored in parallel by executing the QAF on separate quantum cores. It is assured that EnQBCEA performs the search on reduced size text database T, each of size tk. Thus, this increases the success probability for identifying the search results. Lemma 2 states, that to store all tk filtered indices in LAk[tk] we need O((m/C)(tlog2N)) time. Each core QCorec∈QPU utilizes the processing advantage of QMEM in both filtering and searching. Algorithm EnQBCEA-MPM accesses LAk[tk] on each QCorec to obtain the original filtered text indices by applying unitary UkGetL. It takes O(1) time for realizing such transformation under the superposition.

The original text is available in QMEM superposition and shared among all quantum cores. Thus, each QCorec applies QMEM transformation to load all substrings in parallel, such that, each pattern Pk∈P of length Mk(1≤k≤m) verifies for exactness over filtered index approximations. Now, based on the QEM circuit applied under the superposition of tk sized text T, the indices are identified for an exact match. Further, parallel iterations of GSO finds all tk′ (tk′≤tk) solutions of Pk using cth core QCorec in O((m/C)(tk/tk′)) queries. Indeed, the quantum state collapsed while measured; therefore, EnQBCEA repeats GSO operation followed by measurement, on each quantum core QCorec to report all tk′ exact occurrences of the pattern Pk in resulting O((m/C)(tktk′)) queries and thus O((m/C)(tktk′log2tk)) time.

To conclude the complexity, we are considering the maximum reduced size of any filtered location array LAk[tk] as t=tk=max(t1,.,tk,.,tm), and the longest core processing to find maximum pattern occurrences t′=tk′=max(t1′,.,tk′,.,tm′). So, using QPU with C cores and for t′=1 or tfew ( tfew denotes few pattern occurrences ( tfew≪t)), the best case time complexity of EnQBCEA-MPM is O((m/C)(tt′log2t)) and this finds all patterns in parallel. However, when t′=t the worst-case complexity is still bounded to O((m/C)(tlog2t)). A multiplicative factor (log2t) is considered negligible as due to less qubits (tq≪n) needed to expand the reduced search space. However, this factor cannot be ignored when the number of qubits tq is sufficiently large to expand the filtered space (Lomont, 2003; Soni & Rasool, 2021). And the multiplicative constant (m/C) with a small arbitrary constant value of m and constant value of C is found negligible. However, for the comparatively large value of m≫C, a factor m/C cannot be ignored in time complexities. Therefore, quantum search is preferred effectively for few occurrences. Instead, for biological text, t′=t is rare and hence ignored while stating a generalized complexity. We also suggest that algorithm design based on the functionality of GSO, enhances the results with probability Pr[EnQBCEA running at QCorec measures|TiLk⟩QAk in each iteration]≥tk′/tk.

Theoretical Results and Complexities Analysis

The presented algorithms EnQPBEA-MPM and EnQBCEA-MPM are hereby observed with summarized facts of several complexities analysis. This section incorporates the design methods by mainly focusing on actual qubits requirement. For dealing with number of unknown search solutions, the analysis of quantum counting algorithms is included. An idea to simulate QMEM is also discussed here with the realization of quantum effective processing framework.

Summarized complexities analysis and mathematical proved results

We summarize our proven results to compare with the related work. The significant findings were noted herein dedicated tables to emphasize our analytical interpretation. In this section, we present the concluded complexities of our algorithms using Tables 5 and 6 is referred for discussing the design methods with qubits requirement and success probability.

Table 5 Summarized quantum complexities of the proposed algorithms.

Quantum algorithm	Pre-processing complexity	Proposed quantum pattern searching time complexity	Storage complexity (qubits estimation)	
Query	Best time	Worst time	Generalized	
EnQPBEA-MPM	No Text Filtering	O((m/C)Nt)	O((m/C)×(Ntlog2N))	O((m/C)×(Nlog2N))	O((m/C)N)	O((m/C)×(n+Mlog2|Σ|))	
EnQBCEA-MPM	O((m/C)×(tlog2N))	O((m/C)tt′)	O((m/C)×(tt′log2t))	O((m/C)×(tlog2t))	O((m/C)t)	O((m/C)×(n+Mlog2|Σ|+log2t))	

Table 6 Framework and design of proposed algorithms with qubits requirement and success probability.

Quantum algorithm	Algorithm framework	Algorithm design	Quantum registers requirement	Actual qubits requirement	Success probability	
EnQPBEA-MPM	QMEM, QEM, GSO	QPBE Algorithm, Multiple Search Oracle, QPU (C-Quantum Cores)	|Tn⟩QA:m/C, |T[w]⟩QD:m/C, |PM⟩DR: m/C, |q⟩:m/C	(m/C)×(n+ (2(Mlog2|Σ|)+1)	Pr(QCorek)≥ tk/N	
EnQBCEA-MPM	QMEM, QEM, GSO	QBCE Algorithm, Multiple Search Oracle, QPU (C-Quantum Cores)	|Tn⟩QA:m/C, |Tn+1⟩AX:m/C, |T[w]⟩QD:m/C, |Ttq⟩QL:m/C
|PM⟩DR: m/C, |q⟩:m/C	(m/C)×(2n+2(Mlog2|Σ|)+tq+1)	Pr(QCorek) ≥ tk′/tk	

Analysis of proposed algorithms based on several quantum complexities: The resulting complexities of algorithms have been proved earlier and summarized in Table 5. In reference to Tables 1–3, we discuss comparative factors of our work.

Our algorithms obtain speedup with effective quantum processing while comparing with the classical searching time of O(mN) (discussed in the introduction). The classical worst-case time with characters comparison is O(m(NM)) instead, each core of QPU sharing QMEM does parallel match by UkComp, and hence this makes our solutions O((m/C)N) and O((m/C)t) as efficient.

We enhanced the QPBE and QBCE for multiple patterns under quantum architectural and implicit operational parallelism. Based on complexity analysis, our solutions are proved efficient to find an exact match while comparing with existing multiple pattern methods QEMP and QAMP as the factor m cannot be excluded from their complexities.

Our algorithm designs execute exact search oracles in parallel using individual quantum core. The processing time for few pattern occurrences ( tfew≪N) is negligible, and we need less qubits to expand the filtered search space. A QPU runs C cores to search for m/C pattern in parallel, and each quantum core uses its own set of register. So multiplicative constant (m/C) with a small arbitrary constant value of m and constant value of C is found negligible. But, for comparatively large value of m≫C, factor m/C cannot be ignored in time complexities.

We included the storage complexity of our algorithms to estimate the qubits requirement. This complexity is based on the asymptotic estimation of qubits by excluding constants. Later, in Table 6, we specify actual qubits requirement with coefficients to check simulation feasibility of algorithms, as classical machine configuration is restricted to simulate large qubits.

Quantum algorithms and their quantum circuits are proved equivalent for implementations. Therefore, the quantum query, time and storage complexities of proposed algorithms justify their effectiveness. All the requisite and relevant discussions on behalf of Table 5 have been discussed earlier as per the contextual need.

Design methods used for quantum multiple pattern matching algorithms: Table 6 shows framework and design of algorithms. Used quantum registers are mentioned to check for proportional simulation feasibility of actual qubits requirement. Search success probability on any QCorec, proved in theorems, is based on original and filtered text sizes.

The quantum results may contain error while measured; therefore, GSO operation is used to amplify the probability amplitudes. Thus, we obtain the search results with a high probability. Therefore, we categorize our algorithms in BQP complexity class.

The qubits are implicitly analyzed based on the quantum register requirement. For QPU with C cores based parallel processing, this (m/C) factor is there; however, these cores use their own set of quantum registers, so the factor is negligible, and time is reduced in parallelism.

Classical machine configuration is restricted to simulate large qubits and affects simulation. Therefore, algorithms are implemented using hybrid simulation, such that each core can use the sufficient qubits with no excessive increase in qubits requirement of a quantum system.

To save qubits requirement, EnQPBEA and EnQBCEA are simulated by using ANF based quantum operations, dedicated use of ancillary qubits, and utilizing QuEST specific unitary for efficient realization. The hybrid simulation results are noted later in the tables included in “Simulation Results and Discussion” section.

Exact and approximate quantum counting complexity analysis

Quantum counting (QC) is a quantum amplitude estimation method to handle the case of GSO overshooting as t number of search solutions are unknown in advance, so it leads to the unknown number of GSO iterations (Brassard et al., 2002; Nielsen & Chuang, 2010; Song, 2017). These authors (Boyer et al., 1998; Brassard et al., 2002; Lomont, 2003; Younes, 2008) suggested running the quantum counting algorithm initially and then proceeding with the actual number of GSO iterations. We obtained an accurate t value by implementing Exact-QC and the estimated t value through Approx.-QC methods. We provide the complexities analysis of both these cases in the subsection, and the resulting complexities are specified in Table 7.

Table 7 Analysis of QC algorithm used to find approximate or exact value of t as number of search solutions.

Quantum
counting	Algorithm framework	Analyzed complexities for EnQPBEA-MPM	Analyzed Complexities for EnQBCEA-MPM	
Query	Time	Storage	Query	Time	Storage	
Approx. – QC	QAE	O((m/C)N)	O((m/C)×(Nlog2N))	O((m/C)×(n+r)), and (r<n)	O((m/C)t)	O((m/C)×(tlog2t))	O((m/C)×(tq+r)), and (r<tq)	
Exact-QC	QAE	O((m/C)Nt)	O((m/C)×(Ntlog2N))	O((m/C)×(n+r)), and (r=n)	O((m/C)tt′)	O((m/C)×(tt′log2t))	O((m/C)×(tq+r)), and (r=tq)	

Analysis of approximate and exact quantum counting (QC) algorithms: For EnQPBEA algorithm, QC is available with the O(N) query and O(Nlog2N) time. Approx.-QC algorithm can estimate the value of t with some relative error. In contrast, Exact-QC algorithm with O(Nt) query and O(Ntlog2N) time can find the accurate value of t with the high probability (Boyer et al., 1998; Brassard et al., 2002; Lomont, 2003; Younes, 2008). Similarly the EnQBCEA algorithm, working on t filtered indices to find t′ pattern occurrences, needs O(t) query and O(tlog2t) time in Approx.-Q algorithm, and O(tt′) query and O(tt′log2t) time in Exact-QC algorithm (Brassard et al., 2002; Soni & Rasool, 2021). In Table 7, we have shown the complexities by including (m/C) factor because the quantum counting is needed to run on individual quantum core for each pattern separately.

To measure the accurate value of t through Exact-QC we used to take the register with the precision qubits "r"≈(log2N=n) qubits for EnQPBEA and "r"≈(log2t=tq) qubits for EnQBCEA algorithm. Similarly, to measure the approximation of the value of t through Approx.-QC we need a register with precision qubits "r"<log2N qubits for EnQPBEA and "r"<log2t qubits for EnQBCEA algorithm (Brassard et al., 2002; Song, 2017; Soni & Rasool, 2021). The storage complexity showing qubits estimation for Approx.-QC and Exact-QC is also shown additionally in the presented Table 7.

There are two cases to obtain the resulting complexities of combining the QC and GSO as it is further used in our simulation of EnQPBE algorithm – (1) Run Approx.-QC followed by the GSO to find all t occurrences of the pattern, so max(N+Nt)=O(Nt) time; and (2) Run Exact-QC followed by GSO to find all t occurrences of the pattern, therefore Nt+Nt=2×Nt=O(Nt) time. Therefore, the complexity is still bounded by O(Nt) time (Brassard et al., 2002; Lomont, 2003; Song, 2017). Similarly for these cases, the complexity of EnQBCEA algorithm remains O(tt′) time as it works on t filtered indices to find t′ pattern occurrences (Brassard et al., 2002; Soni & Rasool, 2021).

We may expect accurate number of GSO iterations when the exact value of t is obtained through Exact-QC but the deviations in t values are possible through Approx.-QC algorithm, and hence the quantum search results need to be compromised with more errors.

Design and analysis of algebraic normal form to realize QMEM

To simulate our algorithms with the effective quantum processing framework, we propose the design of an algebraic normal form (ANF) circuit for realizing QMEM. Thus, this supports the hybrid simulation (Bogdanova et al., 2018; Hao et al., 2020). We can implement and perform most of the quantum operations directly by utilizing the advantage of ANF that are equivalent to unitary circuits, such as ULoad (QMEM transformation), UComp (QEM operation), and needful quantum adder operation (QAF filtering); hence, this saves the qubits requirement (Malviya & Tiwari, 2020; Malviya & Tiwari, 2021). The other requisite circuits and GSO operations needed for our proposed algorithms will be implemented using combination of ANF and the specific quantum unitary operations available in the QuEST library (defined in next section, used for simulation purpose). For comprehensive understanding of the ANF based QMEM realization refer to Malviya & Tiwari (2020); Soni & Rasool (2021); Soni & Malviya (2021) and Malviya & Tiwari (2021). We proposed a quantum circuit in Fig. 8 showing implicit operational method about the memory processing mentioned in Fig. 2.

Figure 8 Quantum algebraic normal form (ANF) circuit used to realize QMEM processing.

A design of QMEM transformation is proposed here for a main unitary ULoad by using ANF. This will be later used in the next section to simulate QMEM. So, for considered |T2n×w⟩QMEM, the quantum circuit of Fig. 8 creates a superposition of N=2n text addresses by applying H⊗n gates on n qubits address register |Ti⟩QA. These n qubits are used in ANF as n variables to form 2n possible binary strings, usually called Boolean terms. In Fig. 8, the n=4 variables are taken as |T0T1T2T3⟩, where |T0⟩=|23⟩ and |T3⟩=|20⟩ are the most significant and least significant qubit positions. Therefore, the total 24=16 possible terms {|0⟩…,|7⟩,…|F⟩} forms uniform superposition of binary strings {|0000⟩…,|0111⟩,…|1111⟩}.

Further, ANF creates the data superposition, by realizing all the substring data load operation in parallel, each of size M×log2|Σ| in entangled data register |T[i]⟩QD for each |Ti⟩QA. So, for such realization, M×log2|Σ| Boolean functions are computed in parallel, each can have at most O(2n) Boolean terms. These terms are computed with the logical “AND” followed by “XOR” operations. The computation of all possible terms, results output of associated Boolean function. A circuit is shown in Fig. 8 (about Fig. 2) considers pattern of length M=4. Therefore, total 4×2 Boolean functions {|f00⟩,|f01⟩,|f10⟩,|f11⟩,|f20⟩,|f21⟩,|f30⟩,|f31⟩} are computed in parallel as shown in Eqs. (9)–(16) (Bogdanova et al., 2018; Malviya & Tiwari, 2020).

(9) |f00⟩=T2⊕T0T1T2

(10) |f01⟩=T3⊕T2⊕T2T3⊕T1T2⊕T1T2T3⊕T0T2⊕T0T2T3⊕T0T1T3⊕T0T1T2⊕T0T1T2T3

(11) |f10⟩=T3⊕T2⊕T0T1T3⊕T0T1T2

(12) |f11⟩=1⊕T2T3⊕T1T3⊕T1T2T3⊕T0T3⊕T0T2T3⊕T0T1⊕T0T1T2T3

(13) |f20⟩=1⊕T2⊕T0T1⊕T0T1T2

(14) |f21⟩=1⊕T2⊕T2T3⊕T1⊕T1T3⊕T1T2⊕T1T2T3⊕T0⊕T0T3⊕T0T2⊕T0T1T3⊕T0T1T2T3

(15) |f30⟩=1⊕T3⊕T2⊕T0T2T3

(16) |f31⟩=1⊕T3⊕T0T2⊕T0T1⊕T0T1T3⊕T0T1T2

Each function |fxy⟩ with x = [0 to M − 1] = {0, 1, 2, 3} and y = [0 to log2|Σ|] = {0, 1} is computed using variables associated with each term. To load all substring of size M×log2|Σ| in |T[i]⟩QD for each text address |Ti⟩QA in superposition, the binary string equivalent to index position |Ti⟩ is taken as input, and by applying their instances, these Boolean functions are then computed to generate the desired substring within superposition. For example, the loading of text substring indexed at |T4⟩QA, uses binary string |0100⟩QA to load the desired output string in data register |00011011⟩QD (see Fig. 2). This realization facilitates the qubits consuming operation in parallel, and thus, it simulates the quantum algorithms with the minimum qubits requirement (Malviya & Tiwari, 2020; Soni & Rasool, 2021; Soni & Malviya, 2021; Malviya & Tiwari, 2021). The design specifications used for quantum effective processing of algorithms are specified in Table 8 along with interpretation. Later, we will use these designs to simulate our proposed quantum algorithms using the QuEST simulation.

Table 8 Simulation detail to realize the design of quantum effective processing framework.

Simulation of QMEM	Simulation of QEM	Simulation of GSO	
Concept used	Circuit realized	Qubits used	Circuit depth	Concept used	Circuit realized	Qubits used	Circuit depth	Concept used	Circuit realized	Qubits used	Circuit depth	
ANF based circuit (Fig. 8)	UQMEM, ULoad	n+w or tq+w	O(2tq) tq≤n	Boolean Oracle circuit	U Comp	2 × w	O(1)	Phase Oracle circuit	UMark, UDiff	n+1 or tq+1	O(2tq) tq≤n	

Design specifications used for quantum effective processing framework: Table 8 specifies the proposed designs of quantum effective processing framework. It is used in reference with Table 4 to know the quantum gates required for circuit, processing time and qubits needed to realize a circuits.

The ANF circuit is realized as equivalent unitary ULoad (Eq. 3) for QMEM transformation. A circuit implementation needs quantum gates set {H⊗tq,CtqNOT,X⊗tq} with (tq≤n). Such circuit can be simulated later with varying length, as per needed size of QMEM to realize.

The time complexity of ANF based QMEM depends on the circuit depth constructed over tq input variables tq≤n. It realizes M×log2|Σ| Boolean function in parallel, each consist of at most 2tq terms. So, with maximum circuit depth, this circuit will take the exponential complexity O(2tq) as tq≤n under the simulation. We conclude that the physical QMEM processing is remarkable with O(1) time; however, it is exponentially slow in classical.

The O(1) time QEM≈UComp (Eq. 5) design can be simulated efficiently with M×log2|Σ| qubits as all substrings, each of M×log2|Σ| length, can realize in superposition using ANF.

The GSO can be realized as per Table 4. A unitary UMark (Eq. 7) marks index with the phase inversion through (Ctq−1NOT) gate, this flips a target index by inverting ancilla. Further, amplification circuit is used with the set of gates {{H⊗tq,X⊗tq}Ctq−1Z,{X⊗tq,H⊗tq}} (Broda, 2016; Figgatt et al., 2017; Coles, 2020). A circuit depth of GSO is O(2tq), so this depends on text size, as for t sized filtered or N sized original text with t≤N.

Simulation and Experimental Detail

Our proposed algorithms are validated using hybrid (classical and quantum) simulation for the effective realization of equivalent quantum circuits. Therefore, we implemented the algorithms by utilizing the advantage of a C–Library based, flexible simulator with a multi-platform support, called the Quantum Exact Simulation Toolkit (QuEST) (Jones & Benjamin, 2018). We do not intend to analyze the simulation efficiency because of quantum operational restrictions on the classical machine. For a detailed study of QuEST simulation, refer to the published articles (Jones & Benjamin, 2018; Malviya & Tiwari, 2020; Malviya & Tiwari, 2021; Soni & Rasool, 2021; Soni & Malviya, 2021).

QuEST specific simulation features and environment setting

Quantum computations are highly complex, and their efficient simulation on the classical machine is not expected rather than the quantum machine. We performed the simulation to analyze the feasibility of quantum algorithm computations. A quantum machine with a significant amount of qubits still does not exist to realize quantum algorithms. Therefore, we used the QuEST library for the efficient and high-performance simulation of quantum circuits as a substitute for the quantum computer. This simulator is ideal, open-source and available with competent hybrid features such as multithreaded, distributed, and GPU accelerated to use classical hardware for the efficient simulation of quantum circuits. The QuEST simulator proved for the excellent scaling on multicore architectures. Hybrid features of this simulator realized in parallel execution support of OpenMP and MPI. We expect no compromise on simulating the quantum computations even realization is more accurate on a single node, shared memory and distributed systems. A QuES simulation prepares basic and multi-controlled quantum gates as either pure state (vectors) or mixed state (density matrix) under the presence of decoherence (Jones & Benjamin, 2018). This simulation is effective as it performs the quantum operations in the absence of quantum noise.

In their article, Jones & Benjamin (2018), presented the performance comparison of QuEST with the other simulators, and they justified that QuEST is effective because it speeds up simultaneous quantum operations by data parallelism with SIMD execution support. The GPU acceleration is possible through NVIDIA’s CUDA to attain operational speedup and to facilitate parallelism in quantum specific scientific codes. This maintains exponential operations (2n) as the pure quantum state over n–qubits quantum register represented as complex floating-point numbers with default double precision. The quantum multicore realization is the implicit phenomenon that is implemented through the QuEST simulator in the separate quantum execution environment. However, such realization is based on parallel execution of the task in the multithreaded environment over the multiple cores of the CPU (Jones & Benjamin, 2018). Conclusively, we used the QuEST for high-performance simulation of quantum circuits and effective implementation equivalent to quantum algorithms.

We perform the experiments by implementing our quantum algorithms locally on one node with the machine configuration as “Intel i7-7700HQ” processor (having four cores and eight threads) running at 2.80 GHz (having 2400 MHz clock frequency) and 8GB classical RAM (CRAM). We set the QuEST execution environment for either a single or multiple (three) quantum system, each of them contains a separate register with a set of qubits in a pure state to show simulation of our quantum algorithms on single and multicore architecture. The simulation features such as OpenMP is enabled, GPU acceleration is disabled and default double-precision size of 8 bytes is used for reading probability amplitudes; however, the hybrid simulation would be effective.

Description and encoding of biological dataset and patterns

For the simulation purpose, we used the gene sequence database of “Severe Acute Respiratory Syndrome Corona-Virus 2 SARS–CoV–2” for humans. A detailed dataset description and QuEST specific simulation codes are specified within subsection of Additional Information and Declarations entitled “Data and Simulation Codes Availability”.

An idea of implementation is to assign each symbol of the alphabet set Σ with the binary string of length log2|Σ| qubits, and then to transform text database and pattern into binary encoded form. The nucleotide/gene/genome sequence database is preferred for validating our algorithms, so each DNA character of Σ={A,T,C,G} is assigned with the log24=2 length binary string as Σ={00,01,10,11} (Faro & Lecroq, 2009; Soni & Rasool, 2021; Soni & Malviya, 2021). In contrast, the peptide sequences/protein databases with amino acid symbols set |Σ|=20 are ignored here to avoid simulation specific restricted processing of long length binary strings log2|Σ|=5 qubits, as this increases qubits requirement.

The subsets of gene sequence (SARS-CoV-2) is intentionally prepared, as per feasibility of simulation with the text file sizes of {128,256,512} characters. A QPU with C=3 cores is used to realize a case (m≤C) for single pattern |P|=1 and multiple patterns set |P|=3. In case of single pattern, other cores will remain idle. A possible case of (m>C) is shown by taking multiple patterns set |P|=6. For (m=C) case, we take each pattern as of equal size by considering “open reading frame search – patterns” used in the codon process, thus, the pattern of length {3,3,3} characters is taken as {TAA,TAG,TGA} to identify stop codon. And for (m>C) case, the 3 length patterns {TAA,TTT,TAG,TAC,TGA,TGC} are searched using a multicore environment. Next, we take unequal sized patterns {TA,TAG,TGAC} to realize (m=C) case by considering the DNA regular expression based “motif patterns” of length {2,3,4} characters. For (m>C) case, we take the pattern of length {2,2,3,3,4,4} characters as {TA,TC,TAG,TTC,TGAC,TTCA}. The restricted singleton set |P|=1 is used to search single pattern {ATG} (start codon of frame) for existing algorithms. Text and pattern are encoded in binaries, but we specify our results with character file sizes. For each QCorec, we take the pattern of Mk×log2|Σ| qubits, and sequence text of size N=2n indices with word length log2|Σ| qubits. Exact pattern match is performed by exploring text on QMEM (realized by ANF), and by applying QEM circuit for comparing M×log2|Σ| qubits in parallel.

Simulation Results and Discussion

Our proposed algorithms EnQPBEA-MPM and EnQBCEA-MPM were simulated using the QuEST simulator. The experimental results observed during QuEST specific simulation are discussed here in the initial section. In the next section, we suggest some applications related to biological sequence processing for our proposed algorithms.

Simulation detail and analysis with algorithms evaluation criteria

The qubits estimation of a quantum algorithm (or equivalent quantum circuit) shows simulation possibility; however, actual qubits requirement with multiplicative constants decides, whether it is feasible or not. Thus, the performance of QuEST simulation depends on the scaling of multiplicative factors with respect to the data (qubits) processing requirement of quantum circuits. An excessive qubits requirement also limit an underlying configuration of a classical machine. This increases the CRAM workspace and classical CPU processing time with exponential increase. In general, a complete human genome sequence can be excessively large as of (230) nucleotide characters with approximately (3×109) base pairs which are contained in 23 chromosomes, each contains gene sequence of at least (215) DNA/RNA characters (Faro & Lecroq, 2013; Neamatollahi, 2020; Zou et al., 2015). So, for a simulation of n qubits system, QuEST realizes 2n variables (each need 8 bytes of double precision) in O(2n) classical processing time. Therefore, CRAM and a classical CPU processing time proportionally increase as with qubits requirement. For this reason, we prepared the subsets of gene sequence (SARS-CoV-2) with text file sizes of {128,256,512} characters by analyzing the feasible QuEST based hybrid simulation of QPU with C quantum cores accessing text T of size N on shared QMEM.

ANF is actually implemented to simulate the QMEM behaviour and for the other requisite operations. Therefore, in reference to the interpretation of Table 8, we simulate the ANF circuit with varying length and as per needed size of QMEM to realize. However, based on ANF the QuEST simulation of QMEM is observed exponentially slow. The most important point to remember is justified here, that the ANF-based QMEM circuit allows several quantum operations with no increase in qubits requirement. In experimental results, we may observe some deviations and exceptions (if identified) due to implicit random increase in depth of Boolean functions as they are used to simulate QMEM. Further, we use this to perform other requisite quantum operations on the same ANF circuit.

We used two implementations of Boolean oracle circuit for QEM≈UComp (Eq. 5). First, using log2|Σ| sized ancilla qubits to store matching results of each index. Next, we use QuEST specific complex-matrix unitary to find a match and to negate the index for marking. Similarly, the simulation of GSO is realized as per Tables 4 and 8, however, QuEST specific multi-controlled qubit unitary is used to implement the phase inversion.

The QuEST simulator realizes exponential operations effectively by optimizing simulation performance on the classical machine. This simulator provides the log file of quantum assembly instructions (QASM) which help us to record the operations executed on quantum registers by quantum gates and to report execution time of specific quantum circuit during simulation (Jones & Benjamin, 2018; Malviya & Tiwari, 2020; Malviya & Tiwari, 2021; Soni & Malviya, 2021). A CRAM is allocated on demand, so its workspace area may contain several blocks of memory and may be available in compressed form. So, in addition, we used process explorer to measure the maximum workspace requirement of CRAM during the execution of a simulated algorithm.

QuEST specific algorithmic simulation results with observation

This section includes QuEST simulation with observation to map our theoretical – experimental results of algorithms. The results are categorized separately for equal and unequal sized patterns. We noted the results in the tables as per the analysis cases (m≤C) and (m>C) for different text file sizes. A recorded execution log is mentioned in Tables 9 and 10. To prevent the overshooting problem of GSO we implemented exact and approximate quantum counting (QC) algorithms that can find the required number of GSO iterations. So, the observed results of QC and further error analysis are included from Tables 11–13 (for equal sized pattern) and from Tables 14–16 (for unequal sized pattern). The average search time with the memory requirement of the algorithm under QuEST simulation is noted in Tables 17 and 18.

Table 9 Observed outcomes of experimental log during QuEST simulation for equal sized patterns.

Quantum algorithm	Analysis case		Search pattern	Text file size: 128	Text file size: 256	Text file size: 512	
Qubits noted	Quantum gates needed	Qubits noted	Quantum gates needed	Qubits noted	Quantum gates needed	
H	X	R z	CZ	CX	H	X	R z	CZ	CX	H	X	R z	CZ	CX	
EnQPBEA-MPM	(m≤C)	C	P:ATG	17	91	95	141	6	141	18	88	91	265	5	265	19	63	65	545	3	545	
C1	P1:TAA	17	63	65	141	4	142	18	104	105	265	6	266	19	63	63	545	3	546	
C2	P2:TAG	17	63	67	141	4	142	18	136	139	265	8	266	19	117	119	545	6	546	
C3	P3:TGA	17	91	95	141	6	142	18	136	139	265	8	266	19	81	83	545	4	546	
(m>C)	C1	P1:TAA	17	63	65	141	4	142	18	104	105	265	6	266	19	63	63	545	3	546	
C2	P2:TTT	17	21	25	141	1	142	18	56	59	256	3	266	19	45	47	545	2	546	
C3	P3:TAG	17	63	67	141	4	142	18	136	139	265	8	266	19	117	119	545	6	546	
C1	P4:TAC	17	49	52	141	3	142	18	56	58	265	3	266	19	63	64	545	3	546	
C2	P5:TGA	17	91	95	141	6	142	18	136	139	265	8	266	19	81	83	545	4	546	
C3	P6:TGC	17	63	68	141	2	142	18	120	124	265	7	266	19	99	102	545	5	546	
EnQBCEA-MPM	(m≤C)	C	P:ATG	14+5	54	39	308	4	308	16+6	50	36	441	3	442	18+7	53	28	2,366	2	2,366	
C1	P1:TAA	14+5	49	37	598	3	600	16+6	70	56	1,446	4	1,448	18+7	53	37	3,084	2	3,086	
C2	P2:TAG	14+2	24	15	598	2	600	16+3	31	20	1,450	2	1,452	18+4	38	25	3,142	2	3,144	
C3	P3:TGA	14+0	14	6	577	0	578	16+1	19	9	1,169	0	1,171	18+4	38	24	2,785	2	2,787	
(m>C)	C1	P1:TAA	14+5	49	37	598	3	600	16+6	70	56	1,446	4	1,448	18+7	53	37	3,084	2	3,086	
C2	P2:TTT	14+7	49	28	7	2	9	16+8	88	64	8	4	10	18+9	81	54	9	3	11	
C3	P3:TAG	14+2	24	15	598	2	600	16+3	31	20	1,450	2	1,452	18+4	38	25	3,142	2	3,144	
C1	P4:TAC	14+3	39	27	488	2	490	16+4	46	32	1,152	2	1,154	18+5	53	37	3,082	2	3,084	
C2	P5:TGA	14+0	14	6	577	0	578	16+1	19	9	1,169	0	1,171	18+4	38	24	2,785	2	2,787	
C3	P6:TGC	14+1	49	30	483	3	485	16+2	79	56	1,010	4	1,012	18+3	67	42	2,065	3	2,067	

Table 10 Observed outcomes of experimental log during QuEST simulation for unequal sized patterns.

Quantum algorithm	Analysis case		Search pattern	Text file size: 128	Text file size: 256	Text file size: 512	
Qubits noted	Quantum gates needed	Qubits noted	Quantum gates needed	Qubits noted	Quantum gates needed	
H	X	Rz	CZ	CX	H	X	R z	CZ	CX	H	X	R z	CZ	CX	
EnQPBEA-MPM	(m≤C)	C	P:ATG	17	91	95	141	6	141	18	88	91	265	5	265	19	63	65	545	3	545	
C1	P1:TA	15	21	23	141	1	142	16	40	41	265	2	266	17	27	27	545	1	546	
C2	P2:TAG	17	63	67	141	4	142	18	136	139	265	8	266	19	117	119	545	6	546	
C3	P3:TGAC	19	91	96	141	6	142	20	200	204	265	12	266	21	225	228	545	12	546	
(m>C)	C1	P1:TA	15	21	23	141	1	142	16	40	41	265	2	266	17	27	27	545	1	546	
C2	P2:TC	15	35	38	141	2	142	16	40	42	265	2	266	17	45	46	545	2	546	
C3	P3:TAG	17	63	67	141	4	142	18	136	139	265	8	266	19	117	119	545	6	546	
C1	P4:TTC	17	49	53	141	3	142	18	72	75	265	4	266	19	63	65	545	3	546	
C2	P5:TGAC	19	91	96	141	6	142	20	200	204	265	12	266	21	225	228	545	12	546	
C3	P6:TTCA	19	91	95	141	6	142	20	120	123	265	7	266	21	99	101	545	5	546	
EnQBCEA-MPM	(m≤C)	C	P:ATG	14+5	54	39	308	4	308	16+6	50	36	441	3	442	18+7	53	28	2,366	2	2,366	
C1	P1:TA	14+5	29	17	598	1	600	16+6	46	32	1,446	2	1,448	18+7	39	23	3,084	1	3,086	
C2	P2:TAG	14+2	24	15	598	2	600	16+3	31	20	1,450	2	1,452	18+4	38	25	3,142	2	3,144	
C3	P3:TGAC	14+0	14	6	523	0	524	16+0	16	7	1,193	0	1,194	18+0	9	3	861	0	863	
(m>C)	C1	P1:TA	14+5	29	17	598	1	600	16+6	46	32	1,446	2	1,448	18+7	39	23	3,084	1	3,086	
C2	P2:TC	14+5	63	42	435	3	437	16+6	72	48	974	3	976	18+7	81	54	1,955	3	1,957	
C3	P3:TAG	14+2	24	15	598	2	600	16+3	31	20	1,450	2	1,452	18+4	38	25	3,142	2	3,144	
C1	P4:TTC	14+5	77	56	423	4	425	16+6	104	80	896	5	898	18+7	99	72	1,791	4	1,793	
C2	P5:TGAC	14+0	14	6	523	0	524	16+0	16	7	1,193	0	1,194	18+0	9	3	861	0	863	
C2	P6:TTCA	14+3	54	45	443	4	445	16+4	70	55	1,109	4	1,111	18+5	67	50	2,667	3	2,669	

Table 11 Observed results of QC with error analysis in QuEST simulation for N = 128 & equal sized patterns.

Quantum algorithm	Analysis case		Search pattern	Text file size: 128	
Actual patterns	Filtered indices	Error analysis (Exact-QC)	Error Analysis (Approx. –QC)	
Exact QC	No. of CIP	No. of IIP	Error %	Approx. QC	No. of CIP	No. of IIP	Error %	
EnQPBEA-MPM	(m≤C)	C	P:ATG	1	–	1	842	158	15.8	1	836	164	16.4	
C1	P1:TAA	2	–	2	926	74	7.4	1	853	147	14.7	
C2	P2:TAG	2	–	2	923	77	7.7	1	898	102	10.2	
C3	P3:TGA	1	–	1	855	145	14.5	1	846	154	15.4	
(m>C)	C1	P1:TAA	2	–	2	918	82	8.2	1	863	137	13.7	
C2	P2:TTT	10	–	10	850	150	15	11	801	199	19.9	
C3	P3:TAG	2	–	2	919	81	8.1	1	877	123	12.3	
C1	P4:TAC	4	–	4	873	127	12.7	5	804	196	19.6	
C2	P5:TGA	1	–	1	856	144	14.4	1	852	148	14.8	
C3	P6:TGC	2	–	2	912	88	8.8	1	825	175	17.5	
EnQBCEA-MPM	(m≤C)	C	P:ATG	1	3	1	923	77	7.7	1	914	86	8.6	
C1	P1:TAA	2	23	2	953	47	4.7	1	592	408	40.8	
C2	P2:TAG	2	4	2	509	491	49.1	2	501	499	49.9	
C3	P3:TGA	1	1	1	1,000	0	0	1	1,000	0	0	
(m>C)	C1	P1:TAA	2	23	2	969	31	3.1	1	883	117	11.7	
C2	P2:TTT	10	128	10	887	113	11.3	10	867	133	13.3	
C3	P3:TAG	2	4	2	536	464	46.4	2	511	489	48.9	
C1	P4:TAC	4	5	4	736	264	26.4	3	654	346	34.6	
C2	P5:TGA	1	1	1	1,000	0	0	1	1,000	0	0	
C3	P6:TGC	2	2	2	1,000	0	0	2	1,000	0	0	

Table 12 Observed results of QC with error analysis in QuEST simulation for N = 256 and equal sized patterns.

Quantum algorithm	Analysis case		Search pattern	Text file size: 256	
Actual patterns	Filtered indices	Error analysis (Exact-QC)	Error analysis (Approx. -QC)	
Exact QC	No. of CIP	No. of IIP	Error %	Approx. QC	No. of CIP	No. of IIP	Error %	
EnQPBEA-MPM	(m≤C)	C	P:ATG	5	–	5	873	127	12.7	6	818	182	18.2	
C1	P1:TAA	4	–	4	955	45	4.5	6	869	131	13.1	
C2	P2:TAG	2	–	2	944	56	5.6	2	935	65	6.5	
C3	P3:TGA	2	–	2	867	133	13.3	2	867	133	13.3	
(m>C)	C1	P1:TAA	4	–	4	962	38	3.8	6	888	112	11.2	
C2	P2:TTT	16	–	16	886	114	11.4	15	811	189	18.9	
C3	P3:TAG	2	–	2	952	48	4.8	2	971	29	2.9	
C1	P4:TAC	10	–	10	906	94	9.4	10	905	95	9.5	
C2	P5:TGA	2	–	2	888	112	11.2	2	894	106	10.6	
C3	P6:TGC	3	–	3	943	57	5.7	2	901	99	9.9	
EnQBCEA-MPM	(m≤C)	C	P:ATG	5	9	5	925	75	7.5	2	505	495	49.5	
C1	P1:TAA	4	47	4	968	32	3.2	5	915	85	8.5	
C2	P2:TAG	2	7	2	984	16	1.6	2	983	17	1.7	
C3	P3:TGA	2	2	2	1,000	0	0	2	1,000	0	0	
(m>C)	C1	P1:TAA	4	47	4	972	28	2.8	5	909	91	9.1	
C2	P2:TTT	16	256	16	901	99	9.9	15	882	118	11.8	
C3	P3:TAG	2	7	2	978	22	2.2	2	980	20	2	
C1	P4:TAC	10	13	10	852	148	14.8	7	595	405	40.5	
C2	P5:TGA	2	2	2	1,000	0	0	2	1,000	0	0	
C3	P6:TGC	3	3	3	1,000	0	0	3	1,000	0	0	

Table 13 Observed results of QC with error analysis in QuEST simulation for N = 512 and equal sized patterns.

Quantum algorithm	Analysis case		Search pattern	Text file size: 512	
Actual patterns	Filtered indices	Error analysis (Exact-QC)	Error analysis (Approx. -QC)	
Exact QC	No. of CIP	No. of IIP	Error %	Approx. QC	No. of CIP	No. of IIP	Error %	
EnQPBEA-MPM	(m≤C)	C	P:ATG	11	–	11	888	112	11.2	11	881	119	11.9	
C1	P1:TAA	14	–	14	967	33	3.3	15	876	124	12.4	
C2	P2:TAG	4	–	4	978	22	2.2	5	914	86	8.6	
C3	P3:TGA	8	–	8	907	93	9.3	8	900	100	10	
(m>C)	C1	P1:TAA	14	–	14	960	40	4	15	843	157	15.7	
C2	P2:TTT	34	–	34	943	57	5.7	36	907	93	9.3	
C3	P3:TAG	4	–	4	978	22	2.2	5	902	98	9.8	
C1	P4:TAC	15	–	15	934	66	6.6	15	935	65	6.5	
C2	P5:TGA	8	–	8	918	82	8.2	8	900	100	10	
C3	P6:TGC	6	–	6	971	29	2.9	5	917	83	8.3	
EnQBCEA-MPM	(m≤C)	C	P:ATG	11	18	11	974	26	2.6	9	923	77	7.7	
C1	P1:TAA	14	97	14	980	20	2	14	969	31	3.1	
C2	P2:TAG	4	11	4	989	11	1.1	2	545	455	45.5	
C3	P3:TGA	8	10	8	814	186	18.6	7	799	201	20.1	
(m>C)	C1	P1:TAA	14	97	14	987	13	1.3	14	963	37	3.7	
C2	P2:TTT	34	512	34	974	26	2.6	33	928	72	7.2	
C3	P3:TAG	4	11	4	990	10	1	2	506	494	49.4	
C1	P4:TAC	15	19	15	593	407	40.7	15	581	419	41.9	
C2	P5:TGA	8	10	8	805	195	19.5	7	800	200	20	
C3	P6:TGC	6	6	6	988	12	1.2	3	929	71	7.1	

Table 14 Observed results of QC with error analysis in QuEST simulation for N = 128 and unequal sized patterns.

Quantum algorithm	Analysis case		Search pattern	Text file size: 128	
Actual patterns	Filtered indices	Error analysis (Exact-QC)	Error analysis (Approx. -QC)	
Exact QC	No. of CIP	No. of IIP	Error %	Approx. QC	No. of CIP	No. of IIP	Error %	
EnQPBEA-MPM	(m≤C)	C	P:ATG	1	–	1	842	158	15.8	1	836	164	16.4	
C1	P1:TA	10	–	10	579	421	42.1	11	549	451	45.1	
C2	P2:TAG	2	–	2	912	88	8.8	1	892	108	10.8	
C3	P3:TGAC	1	–	1	826	174	17.4	1	814	186	18.6	
(m>C)	C1	P1:TA	10	–	10	596	404	40.4	11	579	421	42.1	
C2	P2:TC	8	–	8	904	96	9.6	11	582	418	41.8	
C3	P3:TAG	2	–	2	924	76	7.6	1	890	110	11	
C1	P4:TTC	3	–	3	891	109	10.9	5	572	428	42.8	
C2	P5:TGAC	1	–	1	833	167	16.7	1	826	174	17.4	
C3	P6:TTCA	1	–	1	855	145	14.5	1	834	166	16.6	
EnQBCEA-MPM	(m≤C)	C	P:ATG	1	3	1	923	77	7.7	1	914	86	8.6	
C1	P1:TA	10	23	10	961	39	3.9	9	957	43	4.3	
C2	P2:TAG	2	4	2	511	489	48.9	2	505	495	49.5	
C3	P3:TGAC	1	1	1	1,000	0	0	1	1,000	0	0	
(m>C)	C1	P1:TA	10	23	10	966	34	3.4	9	957	43	4.3	
C2	P2:TC	8	21	8	943	57	5.7	9	903	97	9.7	
C3	P3:TAG	2	4	2	509	491	49.1	2	503	497	49.7	
C1	P4:TTC	3	19	3	980	20	2	4	961	39	3.9	
C2	P5:TGAC	1	1	1	1,000	0	0	1	1,000	0	0	
C3	P6:TTCA	1	7	1	939	61	6.1	3	854	146	14.6	

Table 15 Observed results of QC with error analysis in QuEST simulation for N = 256 and unequal sized patterns.

Quantum algorithm	Analysis case		Search pattern	Text file size: 256	
Actual patterns	Filtered indices	Error analysis (Exact-QC)	Error Analysis (Approx. -QC)	
Exact QC	No. of CIP	No. of IIP	Error %	Approx. QC	No. of CIP	No. of IIP	Error %	
EnQPBEA-MPM	(m≤C)	C	P:ATG	5	–	5	873	127	12.7	6	818	182	18.2	
C1	P1:TA	19	–	19	665	335	33.5	18	602	398	39.8	
C2	P2:TAG	2	–	2	955	45	4.5	2	952	48	4.8	
C3	P3:TGAC	1	–	1	868	132	13.2	1	861	139	13.9	
(m>C)	C1	P1:TA	19	–	19	617	383	38.3	18	596	404	40.4	
C2	P2:TC	20	–	20	946	54	5.4	22	901	99	9.9	
C3	P3:TAG	2	–	2	961	39	3.9	2	952	48	4.8	
C1	P4:TTC	9	–	9	922	78	7.8	10	907	93	9.3	
C2	P5:TGAC	1	–	1	880	120	12	1	872	128	12.8	
C3	P6:TTCA	3	–	3	910	90	9	2	895	105	10.5	
EnQBCEA-MPM	(m≤C)	C	P:ATG	5	9	5	925	75	7.5	2	505	495	49.5	
C1	P1:TA	19	47	19	975	25	2.5	19	964	36	3.6	
C2	P2:TAG	2	7	2	989	11	1.1	2	973	27	2.7	
C3	P3:TGAC	1	1	1	1,000	0	0	1	1,000	0	0	
(m>C)	C1	P1:TA	19	47	19	975	25	2.5	19	964	36	3.6	
C2	P2:TC	20	44	20	954	46	4.6	19	914	86	8.6	
C3	P3:TAG	2	7	2	962	38	3.8	2	962	38	3.8	
C1	P4:TTC	9	47	9	982	18	1.8	9	977	23	2.3	
C2	P5:TGAC	1	1	1	1,000	0	0	1	1,000	0	0	
C3	P6:TTCA	3	13	3	970	30	3	2	911	89	8.9	

Table 16 Observed results of QC with error analysis in QuEST simulation for N = 512 and unequal sized patterns.

Quantum algorithm	Analysis case		Search pattern	Text file size: 512	
Actual patterns	Filtered indices	Error analysis (Exact-QC)	Error Analysis (Approx.-QC)	
Exact QC	No. of CIP	No. of IIP	Error %	Approx. QC	No. of CIP	No. of IIP	Error %	
EnQPBEA-MPM	(m≤C)	C	P:ATG	11	–	11	888	112	11.2	11	881	119	11.9	
C1	P1:TA	42	–	42	696	304	30.4	43	667	333	33.3	
C2	P2:TAG	4	–	4	970	30	3	5	920	80	8	
C3	P3:TGAC	1	–	1	902	98	9.8	1	904	96	9.6	
(m>C)	C1	P1:TA	42	–	42	623	377	37.7	43	612	388	38.8	
C2	P2:TC	29	–	29	963	37	3.7	30	952	48	4.8	
C3	P3:TAG	4	–	4	975	25	2.5	5	917	83	8.3	
C1	P4:TTC	14	–	14	967	33	3.3	15	935	65	6.5	
C2	P5:TGAC	1	–	1	899	101	10.1	1	900	100	10	
C3	P6:TTCA	5	–	5	945	55	5.5	5	945	55	5.5	
EnQBCEA-MPM	(m≤C)	C	P:ATG	11	18	11	974	26	2.6	9	923	77	7.7	
C1	P1:TA	42	97	42	984	16	1.6	39	911	89	8.9	
C2	P2:TAG	4	11	4	993	7	0.7	2	545	455	45.5	
C3	P3:TGAC	1	1	1	1,000	0	0	1	1,000	0	0	
(m>C)	C1	P1:TA	42	97	42	971	29	2.9	39	900	100	10	
C2	P2:TC	29	71	29	979	21	2.1	28	947	53	5.3	
C3	P3:TAG	4	11	4	975	25	2.5	2	566	434	43.4	
C1	P4:TTC	14	75	14	987	13	1.3	14	989	11	1.1	
C2	P5:TGAC	1	1	1	1,000	0	0	1	1,000	0	0	
C3	P6:TTCA	5	18	5	982	18	1.8	4	966	34	3.4	

Table 17 Experimental realization of algorithms through QuEST specific simulation for equal sized pattern.

Quantum algorithm	Analysis case		Search pattern	Text file size: 128		Text file size: 256		Text file size: 512		
Avg. ET (Sec)	CRAM WS (KiB)	No. of IP	Avg. ET (Sec)	CRAM WS (KiB)	No. of IP	Avg. ET (Sec)	CRAM WS (KiB)	No. of IP	
EnQPBEA-MPM	(m≤C)	C	P:ATG	0.205	5,978	1	0.502	8,702	5	1.409	16,082	11	
C1	P1:TAA	0.132	5,955	2	0.513	8,690	4	1.312	16,087	14	
C2	P2:TAG	0.112	2	0.604	2	2.219	4	
C3	P3:TGA	0.145	1	0.613	2	1.622	8	
(m>C)	C1	P1:TAA	0.233	5,994	2	0.913	8,723	4	3.137	16,116	14	
P4:TAC		4		10		15	
C2	P2:TTT	0.212	10	1.042	16	3.115	34	
P5:TGA		1		2		8	
C3	P3:TAG	0.231	2	1.323	2	4.935	4	
P6:TGC		2		3		6	
EnQBCEA-MPM	(m≤C)	C	P:ATG	0.024	3,661	1	0.099	4,422	5	0.907	7,478	11	
C1	P1:TAA	0.037	4,130	2	0.145	6,447	4	0.901	15,719	14	
C2	P2:TAG	0.027	2	0.126	2	0.888	4	
C3	P3:TGA	0.025	1	0.111	2	0.774	8	
(m>C)	C1	P1:TAA	0.066	4,336	2	0.295	6,592	4	2.051	16,198	14	
P4:TAC		4		10		15	
C2	P2:TTT	0.042	10	0.218	16	1.537	34	
P5:TGA		1		2		8	
C3	P3:TAG	0.052	2	0.244	2	1.675	4	
P6:TGC		2		3		6	

Table 18 Experimental realization of algorithms through QuEST specific simulation for unequal sized pattern.

Quantum algorithm	Analysis case		Search pattern	Text file size: 128		Text file size: 256		Text file size: 512		
Avg. ET (in Sec)	CRAM WS (KiB)	No. of IP	Avg. ET (in Sec)	CRAM WS (KiB)	No. of IP	Avg. ET (in Sec)	CRAM WS (KiB)	No. of IP	
EnQPBEA-MPM	(m≤C)	C	P:ATG	0.205	5,978	1	0.502	8,702	5	1.409	16,082	11	
C1	P1:TA	0.030	11,929	10	0.111	20,903	19	0.331	40,504	42	
C2	P2:TAG	0.112	2	0.618	2	2.232	4	
C3	P3:TGAC	0.468	1	2.404	1	10.436	1	
(m>C)	C1	P1:TA	0.134	12,484	10	0.538	20,960	19	2.058	40,578	42	
P4:TTC		3		9		14	
C2	P2:TC	0.619	8	3.386	20	13.036	29	
P5:TGAC		1		1		1	
C3	P3:TAG	0.717	2	2.832	2	8.976	4	
P6:TTCA		1		3		5	
EnQBCEA-MPM	(m≤C)	C	P:ATG	0.024	3,661	1	0.099	4,422	5	0.907	7,478	11	
C1	P1:TA	0.037	4,130	10	0.147	6,447	19	0.895	15,719	42	
C2	P2:TAG	0.026	2	0.127	2	0.886	4	
C3	P3:TGAC	0.022	1	0.102	1	0.813	1	
(m>C)	C1	P1:TA	0.061	4,242	10	0.264	6,606	19	1.641	16,184	42	
P4:TTC		3		9		14	
C2	P2:TC	0.047	8	0.215	20	1.585	29	
P5:TGAC		1		1		1	
C3	P3:TAG	0.051	2	0.259	2	1.883	4	
P6:TTCA		1		3		5	

Analysis of the experimental log observed during QuEST simulation: For our algorithms EnQPBEA-MPM and EnQBCEA-MPM, Tables 9 and 10 are used to categorize the results between separate text file sizes {128,256,512} and analysis cases (m≤C) and (m>C) are formed for equal – unequal sized multiple pattern set P={P1,P2,P3,P4,P5,P6} of lengths {3,3,3,3,3,3} and {2,2,3,3,4,4}. A QPU with quantum cores C={C1,C2,C3} is considered for the separate execution of desired pattern search.

We assume |P|=1 for ((m=1)<(C=3)) to search for single pattern P of length {3} on |C|=1 i.e. single core to show the simulation of existing QPBE and QBCE algorithms. In this case, other cores are remaining idle. The case of ((m=3)=(C=3)) is considered for searching the desired pattern on individual quantum core. We noted the performance of our algorithms for the case ((m=6)>(C=3)) to realize their executions for large number of patterns (exactly doubled) on constant number of quantum cores. For these cases, we utilized QuEST specific log file that contains a record of standard QASM instructions.

The log record for EnQPBEA and EnQBCEA algorithms are identified for searching, but in case of the EnQBCEA algorithm, the filtering log is additionally recorded. It keeps number of quantum gates needed during the simulation of the quantum algorithm or its equal quantum circuit. A universal quantum gates set {H,X,Rz,Ctq−1Z,CtqX} is noted in QASM log. We used to represent Ctq−1Z and CtqX as CZ and CX to save the text space in tables. The number of coded qubits is observed additionally during a simulation of algorithm within test log.

In reference to Table 6, we simulated the EnQPBEA algorithm using workspace qubits. Instead of actual qubits (n+2×(Mlog2|Σ|)+1), the (n+Mlog2|Σ|+2+2) qubits used in the implementation. Here, all text substrings of size Mlog2|Σ| are realized using ANF, and log2|Σ|=2 workspace qubits store the parallel matching result of each index in superposition. Other two qubits are used as ancillary to support GSO operation. Our findings on qubits, for both the equal and unequal sized patterns, are observed the same as expectations.

The simulation of EnQBCEA algorithm is efficiently coded with (2n+tq) qubits instead of the actual (2n+2×(Mlog2|Σ|)+tq+1) qubits mentioned in Table 6. We took 2n qubits for QAF filtering, and tq qubits to search on filtered indices. All substring of size Mlog2|Σ| are realized using ANF and the pattern is loaded classically. For GSO operation, QuEST unitary complex-matrix is used to find a match and to negate the index for marking. The qubits are observed as the same as expectations for both equal and unequal sized patterns.

We wished to show the implementation using QPU with C quantum cores; thus, the QuEST execution environment was initialized as either a single or multiple (three) quantum system containing a separate register set. We coded hybrid simulation to intentionally save qubits, such that, qubits requirement on any core should not exceed the limits of a classical machine.

Especially for searching, the qubits needed by EnQPBEA is comparatively more than the EnQBCEA because of search is performed with original indices rather than filtered. A QAF takes 2n qubits to filter t indices, so, log2t=tq search qubits may vary as per t value. The expansion of different text search space with reduced indices enhances search mechanism, but this would happen, when a value of t is found as too low as likely the value of log2N=n.

The qubits needed for the searching increases in accordance with the size of the biological text sequence and in direct proportion to the varying length patterns. This phenomenon can be observed in the tables by analysing noted qubits for both equal – unequal sized patterns and text size. A QuEST simulation of algorithms on quantum multicore architecture shows that quantum registers are separately allocated with a set of qubits in the pure state.

We observed the quantum logic gates as implicitly realized under the QuEST simulation of algorithms. For EnQPBEA with equal – unequal sized patterns running on any QCorec. The quantum gates are close proximate values to the gate observed during a single pattern search on a single quantum core. In same context, the number of quantum gates, noted on each QCorec for EnQBCEA, are approximately doubled than single core.

Due to small-sized equal or unequal pattern lengths, the observed number of gates for both the algorithms are analysed in close proximity. However, there is a proportional increase in the gates as with the increase in text file sizes and for the varying length patterns. There exist, huge difference between the gates observation of EnQPBEA and EnQBCEA because the gates observed for EnQBCEA are combined for both filtering and searching. The size of filtered text eventually increases or decreases multiplicity of quantum gates during simulation.

In general, we observed that the simulation takes more quantum gates due to the realization of ANF and other requisite quantum operations. We distributed uniform workload on all cores under the multiple (three) quantum system containing a separate register set. Table 9 shows a case of (m>C) for that the overlapping pattern P2=TTT executed on core C2 takes very less number of gates as due to the reduced depth of ANF circuit. However, we noted proportional increase in the gate counts as per the length and occurrences of search pattern. There is no increase in gate requirements because most of the quantum operations are coded under ANF and this actually saves the specific requirements of quantum gates.

To considering all file sizes, we observed the same growth in the gate counts of Rz and CtqX with a gradual increase. A controlled phase flip gate Ctq−1Z is used to perform the phase inversion on the occurrence identification of pattern over the index. A subset of {H,X} gates are used as per necessity to realize QMEM or diffusion operator of GSO. Any exception other than that are always expected because of the quantum operations are applied over the random increase in depth of Boolean function which is realized in ANF such that tq≤n.

Quantum counting (QC) results and error analysis during QuEST simulation: For our algorithms EnQPBEA-MPM and EnQBCEA-MPM, we categorize the results in tables between separate text file sizes {128,256,512} and the analysis cases (m≤C) and (m>C) formed for equal – unequal sized multiple pattern set P={P1,P2,P3,P4,P5,P6} of lengths {3,3,3,3,3,3} and {2,2,3,3,4,4}. A QPU with quantum cores C={C1,C2,C3} is considered for the separate execution of desired pattern search. We noted the results of equal sized pattern from Tables 11–13 and unequal sized pattern from Tables 14–16.

In reference to the earlier discussions on Grover’s quantum search, initially we implemented our algorithms by assuming that the t number of search solutions (either unique or multiple solution) are already known, and therefore, the GSO iterations were also coded in advance. The case of GSO overshooting is considered as t number of search solutions are unknown. However, it leads to the unknown number of GSO iterations and hence the probability of success would be vanishingly small. So, we handle this by implementing QC algorithm.

To analyze our results, we implemented quantum counting (QC) (Brassard et al., 2002; Nielsen & Chuang, 2010) to estimate the t number of search solutions in advance. We obtained accurate value of t by Exact-QC and estimated value of t through Approx.-QC methods. We know that QC is an amplitude estimation method, therefore, additional quantum register is used with required precision qubits to store the exact or approximate value of t as count. In Exact-QC, we measure the accurate value of t using the register with a precision size ≈ log2N qubits, and we need the register with precision size < log2N qubits to measure the approximate value of t through Approx.-QC (Brassard et al., 2002; Nielsen & Chuang, 2010). So, we coded required qubits in additional register, respectively. After executing Exact-QC and Approx.-QC algorithms, values of t are obtained. And then the algorithm EnQPBEA executes π/4(N/t) and EnQBCEA executes π/4(t/t′) no. of GSO iterations to obtain relative search results. We include the error analysis with the exact value of t Exact-QC and with the approximate value of t (Approx.-QC) in Tables 11–16.

For evaluating the accuracy of search results, we include error analysis with Exact-QC and Approx.-QC cases. So for each pattern, after obtaining the value of t from Exact-QC and Approx.-QC, we repeat EnQPBEA and EnQBCEA algorithms 10 times separately on the individual quantum core. Each repetition completes 100 iterations of algorithms, and after each iteration, we perform the measurement on each core to obtain the search result. Instead of taking the average of 10 times, we have noted the results from Tables 11–16 by taking a summation of 10 repeated executions (each bifurcates the measurement result out of 100 iterations), and hence it is equivalently considered as 1,000 iterations.

We define some requisite parameters which are evaluated for the error analysis purpose, out of 1,000 iterations, such as – (1) No. of Correctly Identified Patterns (CIP): No. of times the pattern identified correctly at the measured index; (2) No. of Incorrectly Identified Patterns (IIP): No. of times the pattern does not found at measured index; (3) No. of Incorrectly Missed Patterns (IMP): No. of times any of the correct pattern index could not be measured; and (4) Error%: ((No.ofIIP/(No. ofCIP+No. ofIIP))×100).

In each of the 10 repeated executions, we coded 100 iterations for sufficient be valuations. If the number of iterations were selected too small ≈25 iterations, then there would have been the chance of getting the No. ofIMP in our evaluations. However, because of the sufficient iterations, we have not reported this case for any search pattern. Therefore, we assure that the likely indices are at least identified during the search phase of both EnQPBEA & EnQBCEA algorithms. We also justify the fact, that the increase in number of iterations also increases the accuracy of measuring all the likely indices. And it also reduces the possibility of pattern that may be incorrectly missed. Similarly, on taking too large number of iterations ≈1,000 iterations, a possibility of getting No. ofIMP will be removed completely, but the algorithm performance becomes worse than the classical equivalent algorithm.

The quantum counting Exact-QC and Approx.-QC are executed 10 times and majority result is considered as correct count of t i.e. either accurate value of t or estimated value of t. The obtained value of t of Exact-QC are found accurate as per actual number of pattern occurrences. As expected, we analyzed the deviations in values of t obtained after executing Approx.-QC algorithm. Therefore, to measure EnQPBEA and EnQBCEA search results, Error% of Approx.-QC case would be comparatively more than the Exact-QC case.

For the case ((m=1)<(C=3)), we show the simulation of existing QPBE and QBCE algorithms. So, a single pattern P of length {3} is searched on a single core with the values of t obtained from Exact-QC and Approx.-QC. In this case, other cores are remaining idle. For the case of ((m=3)=(C=3)) and ((m=6)>(C=3)), we executed Exact-QC and Approx.-QC algorithms on individual quantum cores. After obtaining the separate values of t, the algorithms EnQPBEA and EnQBCEA execute for desired number of search iterations. Evaluating parameters No. ofCIP & No. ofIIP are also evaluated separately on each core. Throughout our experimentation, including exceptional cases, we measured our search results with high probability and with relative Error% value.

We performed the repeated execution of Exact-QC and Approx.-QC for some patterns of equal size {TAA,TAG,TGA} and unequal size {TA,TAG,TGAC} on individual quantum core to analyze (m≤C) and (m>C) cases. So our analysis confirms to obtain the desired values of t on different cores, based on majority, and thus the number of GSO iterations also remains same. However for these cases, based on the evaluating parameters No. ofCIP and No. ofIIP, the resulting outcomes of EnQPBEA and EnQBCEA algorithms were measured with either the similarity or with slight variations.

Practically, based on values of t for Exact-QC and Approx.-QC we coded π/4(N/t) number of GSO iterations in the searching phase of EnQPBEA and EnQBCEA algorithms. There exist some deviations in the estimated value of t through Approx.-QC algorithm. So based on this t value, if π/4(N/t) iterations (rounded off to the nearest integer) remains same as by taking the t value through Exact-QC method, then we identify the same GSO iterations experimentally in both cases. However, the evaluating parameters No. ofCIP and No. ofIIP were measured with either the similarity or with slight variations.

On comparing the results between EnQPBEA and EnQBCEA algorithms, we observed the results of EnQPBEA as consistent and mapped with the theoretical analysis. However, there are two possible factors which are affecting the pattern searching results of EnQBCEA such as – (1) A superposition of the filtered indices (reduced search space of size t≪N) are formed with ⌈log2t⌉=tq qubits and this expands a search space of O(2tq) where tq≤n. Thus, if the indices <2tq then we have a possibility of getting less accurate results as the number of unmarked items are comparatively more in this case. (2) With the reduced search space of size t there exist a possibility of actual pattern occurrences t′≅t/2 (approximately equal to half). In this case, GSO iterations used in EnQBCEA algorithms will realize the problem of balanced function i.e. the pattern occurrence may be checked on the random selection of index from filtered indices. Therefore, the probability of measuring the search result would remain approximately uniform, and it actually generates less accurate results. And in the same exceptional cases, the Error% can also be observed as more.

Based on evaluation parameters No. ofCIP and No. ofIIP the search results of EnQBCEA are obtained well than the EnQPBEA algorithm because of the searching is performed on the filtered indices (reduced search space) rather than the entire available search space which is used by the EnQPBEA algorithm. However, on processing overlapped pattern {P2=TTT} for (m>C) case, we noted the worst outcome of quantum approximate filtering (QAF). Tables 11–13 are showing the improvement in the obtained results. Therefore, in the hypothetical assumption, we may expect the search results of EnQBCEA algorithm with less Error% than the search results of EnQPBEA algorithm.

Analysis of experimental results obtained during QuEST simulation: For our algorithms EnQPBEA-MPM and EnQBCEA-MPM, Tables 17 and 18 are used to categorize the results between separate text file sizes {128,256,512} and analysis cases (m≤C) and (m>C) formed for the equal – unequal sized multiple pattern set P={P1,P2,P3,P4,P5,P6} of lengths {3,3,3,3,3,3} and {2,2,3,3,4,4}. A QPU with quantum cores C={C1,C2,C3} is considered for the separate execution of desired pattern search.

Tables 17 and 18 includes observation on the Avg.ET (Average Execution Time of Searching), CRAM-WS (Classical RAM Workspace), and No. ofIP (Number of Identified Patterns) which are mapped to the observed outcomes of Exact-QC (Exact Quantum Counting) algorithm, see Tables 11–16. We evaluated these parameters for the existing QPBE and QBCE algorithms. So, a single pattern P of length {3} is executed on single core to simulate the case ((m=1)<(C=3)). In this case, other cores are remaining idle.

The case ((m=3)=(C=3)) is considered for searching a desired pattern on individual quantum core. So, Avg.ET is separately noted, but the CRAM workspace is noted for entire execution as the memory is shared among all cores. We used to realize ((m=6)>(C=3)) for the performance evaluation with large no. of patterns (exactly doubled), and these patterns are executed on the constant number of quantum cores.

A Avg.ET was observed using C–Library based clock() function call. It returns several clock ticks since the initiation of QuEST program execution. However, the clock ticks are dependent on processor architecture. So to note a time in seconds, we divide the clock ticks by CLOCKS_PER_SEC. This observation is noted through the test log. A CRAM workspace is observed explicitly by using process explorer to measure the maximum peak of the classical memory throughout the execution of the QuEST program.

Our experiments for the pattern searching was repeated 20 times in a sequence to note their Avg.ET in seconds. The measured time includes the time of quantum superposition realized using ANF to simulate quantum operations in parallel.

Both EnQPBEA and EnQBCEA are found exact for searching the pattern on target indices original or filtered text. The results tested on the dataset within QuEST simulations are here validated. Algorithms identify all pattern occurrences with high probability and in less time of execution. However, the search results of EnQBCEA are found optimal due to the search is performed on filtered space of size t rather than the original space of size N. Even on a single core, these results are optimized because of the same pattern is searched over the filtered text.

The algorithms' performance observed in proportional increase with Avg.ET of searching, concerning the increase in text file sizes. We have stated earlier that our intentions are not to analyze the simulation efficiency due to performance restrictions on the classical machine. However, we ensure that for our text file sizes and patterns the time needed by a real quantum machine will be negligible. Average times noted for algorithms are specified explicitly for each core; but, due to parallel realization on the quantum multicore concept, we consider a maximum time taken by any core among C-QCore.

Due to small-sized equal or unequal pattern lengths, the Avg.ET observed for both these algorithms are analyzed in close proximity. However, all the occurrences of each pattern are reported either within the original or filtered text sequence (see tables). For a case of (m>C) we distributed uniform workload on all the quantum cores under the multiple (three) quantum system containing a separate register set. Tables 17 and 18 shows proportional increase in the Avg.ET values as per the increase in file sizes. And in the same case, Avg.ET of the EnQBCEA algorithm is found optimal than the EnQPBEA algorithm.

The search time is dependent on the size of the text sequence and the number of occurrences to report for each pattern; therefore, we consider slight deviations. For all file sizes, and the equal or unequal sized patterns, we noted the Avg.ET on individual cores. Here, the time is deviating in accordance with the size and frequency of pattern occurrences within the text sequence. Some exceptions are considered here because of implicit random increase in depth of Boolean functions used in ANF based hybrid simulation. Recall, such an implementation aspect gives us privilege to save the number of qubits required for a simulation of algorithms.

We restate that algorithmic performance on simulation may affect due to the scaling factors associated with qubits; thus, this also increases the workspace requirement of CRAM and processing time with an exponential increase. Memory requirement is also a crucial cum critical factor that may limit the execution of QuEST specific simulated program. So, we prepared a very small-sized data processing requirement of text and pattern and observed the utilization of the CRAM workspace (in KiB) throughout the execution of QuEST program.

In reference to Avg.ET, we noted workspace utilization of CRAM. Therefore, the specified workspace in Tables 17 and 18 shows the average of repetitive experiments that were performed 20 times. A CRAM consumption is observed separately with respect to single pattern on single core. We noted the combined workspace for the cases (m=C) and (m>C) to search for multiple string patterns, each one runs on separate quantum core.

This is observed throughout the execution of algorithms that, the CRAM consumption of a single core is less on comparing with multiple quantum cores sharing. We expect this under QuEST simulation because the execution environment was set to a single quantum system with assigned registers to realize a single quantum core. For EnQPBEA and EnQBCEA, and (m=C) case, the execution environment of QuEST was set to multiple quantum systems with their separate registers of needed qubits to realize multiple quantum cores as a simulation of physical quantum multicore machine. So, cross-comparison assures that CRAM workspace is usually more. Similarly, for (m>C) case, we are observing the expected increase in CRAM workspace as each quantum system can simulate the individual quantum core to execute EnQPBEA and EnQBCEA algorithms twice to complete the execution.

A CRAM workspace will gradually increase with respect to the text file sizes. Thus, this proportional phenomenon may restrict the classical simulation of quantum behaviour for processing the large sequence databases, usually of at least exponential in size. As well as, to process a large number of multiple string pattern m on the small number of available quantum cores C, there would be an eventual increase in the size of CRAM utilization. For all the cases (m<C), (m=C) and (m>C) our Tables 17 and 18 shows the proportional increase in the CRAM workspace values as per the increase in file sizes. And for same cases, CRAM workspace of the EnQBCEA algorithm is found optimal than the EnQPBEA algorithm.

Since we observed that the CRAM utilization of EnQBCEA for their equal – unequal sized patterns are found in the close proximate regions. However, there exists much more difference in the CRAM consumptions of EnQPBEA due to the reported pattern occurrences over the original text and implicit random increase in depth of Boolean functions used in ANF.

Our observations on QuEST specific simulation mainly involves the critical factor of qubits requirement for simulating quantum algorithm. It may cause exhaustive use of CRAM, and the classical CPU computation time is also increased with the at least exponential factor to process the circuit depth of quantum algorithms. However, we implemented quantum algorithms with hybrid simulation by effectively utilizing QuEST performance with several optimizations.

Proposed algorithmic applications to process biological sequences

This section defines several applications of our proposed quantum algorithms related to search multiple patterns within the biological sequence databases. Table 19 specifies the applicability of proposed algorithms with respect to significant characteristics and performance restrictions.

Table 19 Applications specific detail of proposed algorithm to process biological sequences.

Quantum algorithm	Significant characteristics	Performance restrictions	Biological sequence and databases	Specific applications	
EnQPBEA-MPM	* Suitable for processing multiple patterns in an effective manner as its design utilizes multiple cores to search for Pk on shared QMEM.
* This performs exact search, thus it is more practicable for processing biological sequences efficiently.
* All exact occurrence of each Pk are found through QCorec of QPU having C cores in O((m/C)N).
* Suitable to search for long length patterns either formed over |Σ|=4 (DNA) or |Σ|=20 (Amino Acid), as match takes O(1) on QMEM.
* Exponential sized text sequence is effectively search for each pattern, irrespective of text size & frequent pattern occurrence with speedup.
* Sets benchmark to find multiple pattern using multicore parallelism on text with Pr(QCorec)≥ tk/N.	* High probable search results may be affected on each QCorec while processing exponentially large size text with few Pk occurrences.
* For large alphabet set Σ such as |Σ|=20 (Amino Acid), the qubits requirement is excessively high, as of now, it is restricted, however, no limitation on quantum machine.
* Search time is still dependent on cth core QCorec, so, core running for the unequal sized pattern with expected more frequent occurrence, degrades algorithm performance.
* The average probability of search result, with N sized text & t marked index, are proportionally increased with successive measurements.
* A O(2n) depth ANF circuit slows down the simulation, and thus, this affects individual QCorec output.	* DNA/RNA text is searched with a long length pattern. Equal and unequal pattern length is preferred on genome sequence. A sequence database for such examples are GenBank, DDBJ, EMBL.
* Search for multiple amino acid pattern in protein database with prefer able moderate length patterns. This reduces the searching overhead. Example of some database are the GenBank, DDBJ, EMBL, GenPept, PROSITE, Swiss-Prot.	* DNA/RNA/Genome/Protein sequencing.
* Local and the global sequence alignments techniques, similarity detection.
* Gene and genome analysis, mapping and comparison with other similar genes of same/different organisms.
* The DNA mutation, compare investigated DNA with the known sequence.
* Motif finding, open reading frame search and codons matching/recognition.
* The proteogenomics mapping read maps on genomic sequence.	
EnQBCEA-MPM	* Performs multiple patterns search on filtered text in effectively as its design utilizes the multiple cores to search for Pk on shared QMEM.
* All exact occurrence of each Pk are found through QCorec of QPU having C cores in O((m/C)t).
* Exact matching is preferred over large text that may contain frequent pattern occurrence, thus, significant to process a biological sequence.
* Search mechanism is effective as because of finding patterns over the reduced size text, instead original.
* This algorithm is remarkable over all classical and especially existing quantum multi-pattern methods.
* Each core assures to report pattern match with Pr(QCorec)≥tk′/tk over individual filtered text indices.	* The probability of search results at kth core QCorec will depend on relativeness of individual filtered indices to the occurrences of pattern present in filtered text for each Pk.
* Bothe filtering and search time is still dependent on cth core QCorec, so, core running for unequal sized pattern with more filtering outcome and frequent search occurrence may degrades algorithm performance.
* Due to algorithmic filtering, the qubits requirement increases with (m/C), thus, restricts simulation.
* Performance on each QCorec is affected with unequal length pattern and its formation over large |Σ|.
* O(2tq) tq≤n ANF circuit depth slows down simulation, and thus, it affects individual QCorec output.	* Multiple codon can code for same amino acid with either single or the multi locations within sequence.
* DNA/RNA/Peptide & Protein sequences are preferably search with the small length pattern for simulation and no restrictions on quantum machine.
* The biological text sequence database as can search for multi pattern. In example GenBank, Nucleotide database, PROSITE, GenPept, Swiss-Prot, DDBJ, EMBL.	* DNA/RNA/Genome/Protein sequencing.
* Preferable approach for method of multiple sequence alignment.
* Motif finding, open reading frame search and codons matching with using a similarity detection/checking.
* Apply over specific nucleotide or peptide sequences to deal with the local alignment.
* Apply to a sequence alignment (global) on genome or protein.
* Applicable on gene mapping and the exact substring matching.	

In Table 19, we summarize the significant characteristics and performance restrictions of the presented algorithms. We highlighted main points with respect to the contextual interpretation of biological text sequences and their standardized databases. To have more understanding of the algorithms, we direct the reader to specific applications (Sheik, Aggarwal & Anindya Poddar, 2004; Basel, 2006; Choo, 2006; Kalsi, Peltola & Tarhio, 2008; Fredriksson, 2009; Charalampos, Panagiotis & Konstantinos, 2011; Rivals, Salmela & Tarhio, 2011; Faro & Lecroq, 2013; Jiang, Zhang & Zhang, 2013; Singh, 2015; Zhang et al., 2015; Tahir, Sardaraz & Ikram, 2017; Hakak & Kamsin, 2019; Neamatollahi, 2020; Soni & Rasool, 2021; Soni & Malviya, 2021; Raja & Srinivasulu Reddy, 2019). These articles are related to process biological sequences and their databases.

In general, we say that the presented algorithms to process biological sequences, are influenced by three parameters such as alphabet size, pattern length and the size of the text. These parameters may affect the performance of the algorithmic simulation. However, their realization of quantum machines would be effective in specific biological applications.

The probability of search results is based on the relativity between pattern occurrences and the size of the text database (original or filtered). Therefore, the search results are obtained in the best time with at least half probability, and for more frequent pattern occurrences, the results are obtained in the worst time with very high probability.

In multiple pattern processing, there exist some variations in the performance of algorithm. It is because of processing equal or unequal size patterns. The simulation over a very large-sized biological sequence database is not feasible for simulation because of higher qubits requirement; therefore, a subset of the database is searched for a pattern as per the feasibility. There is no such restriction on real quantum machines as they can realize effective processing.

Conclusion and Future Work

In this work, we enhanced the existing quantum pattern matching methods QPBE and QBCE to search multiple patterns in parallel by using QPU with C cores accessing text on shared QMEM. The search time to find all occurrences of the individual patterns overlapped implicitly. Based on several complexity analysis factors, our proposed quantum algorithms EnQPBEA-MPM and EnQBCEA-MPM are proved efficient to find exact patterns while comparing with existing multiple pattern methods such as QEMP and QAMP as their quantum design cannot exclude multiplicative factor m. A design of presented algorithms uses architectural parallelism, but with a multiplicative constant m/C. This factor can be negligible for small arbitrary constant value of m and constant value of C. However, for comparatively large value of m≫C, a factor m/C cannot be ignored in the time complexities. Similarly, due to an implicit operational parallelism, the logarithmic factor is found negligible when the original or filtered search space remains too small to expand in superposition. However, this logarithmic factor cannot be ignored with large number of qubits. Indeed, our proposed algorithms are preferred effectively for finding the few pattern occurrences. Therefore, to process the exponentially large size biological text sequences, our O((m/C)N) and O((m/C)t) time quantum solutions are efficient, and they outperform over existing classical as well as quantum solutions by achieving speedups. The algorithms are justified, based on mathematical proves, as equivalent to quantum circuits. To obtain the accurate search results, quantum counting is explicitly added to the functionality of proposed algorithms. We suggested specific applications of these algorithms related to biological sequence processing.

The quantum algorithms are validated through restricted simulation performance. We used Exact-QC to measure exact value of t and to validate the accurate search results. However, we analyzed the deviations and less accurate search results by combining Approx.-QC and GSO operator. The possible cases (m≤C) and (m>C) were used in our experimentation to observe the variations in search results. Indeed, our intentions were not to analyze the simulation efficiency; therefore, as per the feasibility, we presented the hybrid simulation to realize quantum operations of the algorithm on the classical machine. However, we seek their efficient execution on the real quantum machine to observe the high-performance computation aspects. Further, the proposed work can be extended possibly either to replace filtering approximations of EnQBCEA with exactness or to modify this using other error metric methods to increase accuracy. The open problems would be the realizations of multiple oracles in parallel on a single quantum core, such that the multiplicative factors can be completely removed, and the design of search method through phase matching as replacement of amplitude amplification.

Supplemental Information

Supplemental Information 1 Appendix A: List of Abbreviations.

Click here for additional data file.

Supplemental Information 2 Appendix B: Nomenclatures used in Proposed Algorithms: EnQPBEA & EnQBCEA.

Click here for additional data file.

Supplemental Information 3 Appendix C: Correctness Proof of Proposed Algorithm 1: EnQPBEA-MPM.

Click here for additional data file.

Supplemental Information 4 Appendix D: Correctness Proof of Proposed Algorithm 2: EnQBCEA-MPM.

Click here for additional data file.

The authors are thankful to the domain researchers for sharing their ideas to extend over upcoming quantum technology. We are also thankful to the reviewers as their valuable suggestions improved the quality of the article.

Additional Information and Declarations

Competing Interests

Author Contributions

Data Availability

The authors declare that they have no competing interests.

Kapil Kumar Soni conceived and designed the experiments, performed the experiments, analyzed the data, performed the computation work, prepared figures and/or tables, authored or reviewed drafts of the paper, quantum algorithm design & analysis, and writing of complexity proofs, and approved the final draft.

Akhtar Rasool analyzed the data, authored or reviewed drafts of the paper, and approved the final draft.

The following information was supplied regarding data availability:

The sequences are available at Genbank: MW687138.

The QuEST Simulation Codes are available at GitHub:

https://github.com/profkapilsoni/EnQPBEA-and-EnQBCEA-Algorithms.

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
