# Peer review of "Quantum-effective exact multiple patterns matching algorithms for biological sequences"

_PeerJ Computer Science, doi:10.7717/peerj-cs.957_

## Round 0.1 · original submission · Major Revisions

Please incorporate the comments of the reviewers.

Reviewer 1 ·

Basic reporting

Avoid using abbreviations in abstract unless it’s a time complexity.
Please provide a table for all the abbreviations.
Please merge the Prior work and Important Findings and Related work section into one section.

Experimental design

Theorem 1 and 2 are defined twice. It should be defined once only in the proposed methodology section.

Validity of the findings

The Section- ‘Simulation detail and analysis with algorithms evaluation criteria’ should be added as a subsection within Results and Discussions.

Reviewer 2 ·

Basic reporting

The article claims to proposes an efficient quantum solutions for exact multiple pattern matching to process the biological sequences. It appears that a lot of effort has been made in writing the article for presenting the solution through complexity analysis of algorithm. Pl. refer to the attachment.

Experimental design

Experimental Design is weak due to limitation of current hardware technology so cannot be basis of a publication. Further, Grover's search can overshoot if the number of solutions are not known in advance, which is going to be the case when t exact pattern matches are not known, so how are you going to handle such cases while performing simulation. Ideally there should be table for depicting numerical values of variables like N, m, t etc. along with constants C etc. and performances in time and space rather than equations. Further Table 11 and 12 should also report on the number or percentage of patterns correctly identified, incorrectly identified and incorrectly missed etc.

Validity of the findings

The article claims to proposes an efficient quantum solutions for exact multiple pattern matching to process the biological sequences. It appears that a lot of effort has been made in writing the article for presenting the solution through complexity analysis of algorithm. The article claims to find all

Additional comments

This work should be divided into two parts viz., theoretical and application part with simulations giving details on experimental accuracy of the proposed algorithm.

Annotated reviews are not available for download in order to protect the identity of reviewers who chose to remain anonymous.

---

## Round 0.2 · Minor Revisions

You are required to address the concerns of the reviewer.

Reviewer 1 ·

Basic reporting

1.All the previous concerns have been addressed except usage of abbreviations in abstract.
2. English should be corrected.Grammatical errors like "At present, there is no effective quantum solution exists to 26 process multiple patterns." should be corrected.
3.In "Motivation and contribution of work" a point is given as follows:
Existing

Experimental design

1.In Proof of Theorem 1 what does 't' stand for.

Validity of the findings

In the Results Section, observations are given Table wise, instead the findings that they emphasize should be given as headings and relevant Tables should be included under them.

---

## Round 0.3 · accepted · Accept

Congratulations on the acceptance of your paper.

Reviewer 1 ·

Basic reporting

All previous concerns have been addressed.
A table containing all abbreviations should be added.

Experimental design

The corrections have been done as per requirement.

Validity of the findings

The impact and novelty has been properly presented